# A Raman Lidar Tropospheric Water Vapour Climatology and Height-Resolved Trend Analysis over Payerne Switzerland

Shannon Hicks-Jalali[1], Robert J. Sica[1,2], Giovanni Martucci[2], Eliane Maillard Barras[2], Jordan Voirin[3,2], and Alexander Haefele[2,1]

[1]Department of Physics and Astronomy, The University of Western Ontario, London, Canada
[2]Federal Office of Meteorology and Climatology MeteoSwiss, Payerne, Switzerland
[3]Triform SA, Fribourg, Switzerland

*Correspondence to:* Shannon Hicks-Jalali (shicks26@uwo.ca)

**Abstract.**

Water vapour is the strongest greenhouse gas in our atmosphere and its strength and its dependence on temperature leads to a strong feedback mechanism in both the troposphere and the stratosphere. Raman water vapour lidars can be used to make high vertical resolution measurements, on the order of 10s of meters, making height-resolved trend analyses possible. Raman water vapour lidars have not typically been used for trends analyses, primarily due to the lack of long enough time series. However, the RAman Lidar for Meteorological Observations (RALMO), located in Payerne, Switzerland, is capable of making operational water vapour measurements and has one of the longest ground-based and well-characterized data sets available. We have calculated a 11.5-year water vapour climatology using RALMO measurements in the troposphere. Our study uses nighttime measurements during mostly clear conditions, which creates a natural selection bias. The climatology shows that the highest water vapour specific humidity concentrations are in the summer months, and the lowest in the winter months. We have also calculated the geophysical variability of water vapour. The percent variability of water vapour in the free troposphere is larger than in the boundary layer.

We have also determined water vapour trends from 2009 - 2019. We first calculate precipitable water vapour trends for comparison with the majority of water vapour trend studies. We detect a nighttime precipitable water vapour trend of 1.3 mm/decade using RALMO measurements, which is significant at the 90% level. The trend is consistent with a 1.38°C/decade surface temperature trend detected by coincident radiosonde measurements under the assumption that relative humidity remains constant;however, it is larger than previous water vapour trend values. We compare the nighttime RALMO PWV trend to daytime and nighttime PWV trends using operational radiosonde measurements and find them to agree with each other. We cannot detect a bias between the daytime and nighttime trends due to the large uncertainties in the trends. For the first time, we show height-resolved increases in water vapour through the troposphere. We detect positive tropospheric water vapour trends ranging from 5% change in specific humidity per decade to 15% specific humidity per decade depending on the altitude. The water vapour trends at 5 layers are statistically significant at or above the 90% level.

# 1 Introduction

Water vapour is the atmosphere's most important minor constituent. It plays a significant role in almost all aspects of the atmosphere, including: dynamics, circulation, radiative processes, and chemistry with other molecular and particulate species. Water vapour accounts for 60% of the greenhouse effect for clear skies (Kiehl and Trenberth, 1997) and has a significant

feedback response to increases in temperature. It is, in this sense, Earth's "strongest" natural greenhouse gas. In fact, the temperature increase due to a doubling in $CO_2$ would be magnified to twice its predicted change from $CO_2$ alone if water vapour feedback is included in global climate (Held and Soden, 2000). While the majority of water vapour feedback is due to infrared absorption in the upper troposphere and lower stratosphere (UTLS), the change in lower and mid-tropospheric water vapour also has large implications for the hydrologic cycle and water vapour feedback (Dessler et al., 2013; Held and

Soden, 2006). Tropospheric water vapour plays a key and complicated role in the hydrologic cycle, and it remains unclear how increases in water vapour affects precipitation rates, the chance of extreme precipitation events, or droughts. High vertical resolution water vapour profiles on the order of $100\,m$ are extremely useful for modelers to accurately reproduce convection (Weckwerth et al., 1999). Tropospheric water vapour climatologies and trends are also critical to understanding and modeling water vapour's impact on global circulation systems. Tropospheric water vapour is predicted to increase by 7% for every $1°C$

increase in temperature in the lower troposphere and at even higher percentages (up to 15%) in the UTLS under the assumption that relative humidity is conserved (Held and Soden, 2000; Sherwood et al., 2010).

Measuring an atmospheric water vapour trend is difficult due to the fact that the magnitude of the water vapour concentration at a given altitude in the troposphere can change by more than 100% on a daily basis. Due to this huge variation, quantifying water vapour trends requires a level of precision and consistency in the measurements which is practically difficult to achieve.

Early studies of water vapour trends in the troposphere were typically conducted with radiosondes (Hense et al., 1988; Ross and Elliott, 1996; Ross and Elliot, 2001). However, the trends calculated in those studies were most often not significant and their magnitudes varied greatly. Radiosonde biases have now been well documented and data sets require significant corrections and homogenization in order to be used (Elliott and Gaffen, 1991; McCarthy et al., 2009; Miloshevich et al., 2009). The GCOS Reference Upper Air Network (GRUAN) has recently made a significant step in homogenizing and correcting many modern

radiosondes so that they can be more reliably integrated into trend analyses (Miloshevich et al., 2009; Immler et al., 2010; Dirksen et al., 2014).

Since radiosondes can require significant corrections, water vapour trends and climatology studies moved towards including other instruments such as microwave radiometers (Morland et al., 2009; Hocke et al., 2011), satellite-based instruments such as the microwave limb sounders (Hegglin et al., 2014; Khosrawi et al., 2018), global positioning systems (GPS, Jin et al.

(2007); Wang et al. (2016)), and reanalyses models (Trenberth et al., 2005). However, the majority of these instruments can only measure columns of water vapour, or partial column profiles. Most satellite-based instruments are capable of measuring water vapour down to $300\,hPa$, or roughly $9\,km$ altitude (Hegglin et al., 2013), but a few like the AIRS instrument on the Aqua satellite can accurately measure through the troposphere (Trent et al., 2019). However, while satellite measurements provide excellent global coverage, their vertical resolutions are typically on the order of kilometers which limits their ability

to capture water vapour's large variability with altitude. Unlike these instruments, a Raman water vapour lidar has the ability to measure high resolution water vapour profiles in the troposphere (on the order of meters), as well as calculate column measurements (Melfi, 1972; Whiteman, 2003). Lidars' high vertical resolution compared to microwave limb sounders, or microwave radiometers, make them ideal instruments for studying the evolution of water vapour in the troposphere. The major drawbacks to using Raman lidars for trend measurements are that they are difficult to fully automate, and they can not measure during precipitation events or in thick cloudy conditions. As such, there have not been many lidars which have long enough data sets with enough stability to detect statistically significant trends for climatological studies. As far as we are aware, there have been only four publications on operational Raman water vapour lidars in the last 2 decades (Goldsmith et al., 1994; Dinoev et al., 2013; Hadad et al., 2018; Reichardt et al., 2012). While these lidars have been run operationally over the last 2 decades, RALMO is the only one which has presented a water vapour trends study which we show here.

Many studies have been made of precipitable water vapour (PWV) trends across Europe and Switzerland. Ross and Elliot (2001) calculated PWV trends from 1958–1995 using radiosonde measurements from the surface to 500 hPa and found highly geographically variable trends around the globe. Their trends calculated across Europe were insignificant and differed in sign depending on the location. Trenberth et al. (2005) also calculated global PWV trends using radiosondes and models over the ocean and found mean ocean trends of 1.6 mm/decade using measurements from 1989 to 2004. Morland et al. (2009) calculated PWV trends using a radiometer, radiosonde, and model measurements in Bern, Switzerland from 1996 to 2007. Only the radiosonde measured a statistically significant trend at night of 0.49 mm/decade. Hocke et al. (2011) published an updated trend using homogenized measurements from 2 radiometers from 1994–2009. They did not detect an overall trend, but found seasonal trends of 10% change in PWV per decade in the summer and -15% per decade in the winter. Other studies around Europe using measurements from the 1980s up through 2008 report positive trends on the order of 0.3–0.5 mm/decade (Nilsson and Elgered, 2008; Ning and Elgered, 2018). Hadad et al. (2018),hereafter H2018, also measured PWV trends using ground-based GPS measurements and found a positive trend but insignificant trend of $0.42 \pm 0.45$ mm/decade. Recently, a water vapour lidar climatology using Raman lidar measurements was published by H2018 for measurements over France (45.75°N, 3.125°E) from 2010 to 2016. The measurements in our study have 5 more years of measurements compared to H2018. H2018 also conducted a water vapour trends analysis for their site using satellite, GPS, and ground-station measurements as well as ERA-Interim reanalyses; however, they did not include lidar measurements in their trend analysis.

While there are many PWV trend analyses, there appears to be a large gap in the community in the area of height-resolved trends. The lack of height-resolved trends may be because there are very few instruments routinely operated with the capability to measure atmospheric profiles at resolutions under 1 km in the troposphere (Wulfmeyer et al., 2015). Some studies have measured trends at the surface as well as 850 hPa using radiosondes and models (Hadad et al., 2018; Serreze et al., 2012). H2018 also uses measurements from the AIRS satellite to detect trends and calculates a water vapour mixing ratio trend of 0.13 g/kg per decade at 950 hPa and at 850 hPa. To our best knowledge, ours is the first study to use a Raman lidar to calculate water vapour trends at different layers in the troposphere above 850 hPa.

This study presents a tropospheric water vapour climatology and trend analysis using 11.5 years of nighttime measurements from the RAman Lidar for Meteorological Observation (RALMO) over Payerne, Switzerland. While not the first study to

produce a tropospheric water vapour climatology from a Raman lidar, it is the first to use Raman lidar measurements to calculate precipitable water vapour (PWV) trends, as well as trends at different layers in the troposphere. Section 2 will discuss the measurements used in this study. Section 3 describes the methodology used to create the climatology and the calculation of geophysical variability over the ten years. Section 4 presents the trend results for both PWV and specific humidity at ten different pressure levels up to 250 hPa. Sections 5 and 6 present the Discussion and Summary and Conclusions of our study, respectively.

## 2 Description of Instruments and Measurements

### 2.1 Raman Lidar

The lidar measurements were taken by RALMO in Payerne, Switzerland ($46.81°$N, $6.94°$E, 491 m a.s.l.). RALMO is a fully-automated operational water vapour lidar capable of reaching into the upper troposphere at nighttime and has been designed to run autonomously with minimal downtime, high accuracy, and temporal measurement-stability (Dinoev et al., 2013; Brocard et al., 2013b). Over the last ten years RALMO has been operating both day and night with an average of 50% uptime, with 40% of the downtime due to precipitation or the presence of clouds below 900 m. The other 10% of the downtime is due to routine maintenance and equipment malfunctions. RALMO uses a tripled ND:YaG laser operating at 30 Hz with a pulse power of 300 mJ. The laser has the ability to operate at 450 mJ, however, 300 mJ was chosen to maximize the lifetime of the flashlamps. RALMO includes 12 detection channels: 4 elastic or Rayleigh-scatter channels (near- and far-field), 4 pure rotational Raman, 2 Raman nitrogen (386.7 nm, digital and analog), and 2 water vapour (407.45 nm, digital and analog) channels. Raw measurements are recorded as 1800-shot profiles (approximately 1 minute) with an altitude resolution of 3.75 m for each channel from the surface up to 60 km. The raw 1 min profiles are called "scans". This work uses measurements from the pure rotational Raman channels and elastic channels to calculate nightly aerosol backscatter ratios, as well as measurements from the water vapour and nitrogen vibrational-rotational Raman channels to retrieve the water vapour mixing ratio profiles used in the climatology. The aerosol backscatter ratios are used to calculate transmissions and aerosol extinction for the water vapour retrieval.

The specific humidity profiles used for the climatology and trends analysis are determined from the RALMO measurements using the optimal estimation method(OEM) retrieval introduced in Sica and Haefele (2016) for water vapour lidars. While the OEM retrieval is in units of water vapour mixing ratio, it is finally converted to specific humidity for the purposes of this trend analysis to maintain consistency with the water vapour trend literature. The OEM uses Bayes' theorem to constrain the solution space for the retrieval. It does this by adding in the use of an *a priori* state ($\mathbf{x_a}$). A probability of any given state of the system is assigned, assuming the errors of the system are Gaussian. The optimal solution for the system is then found by minimizing the cost of the solution, where the cost is defined as:

$$Cost = \left[\frac{1}{2}(\mathbf{y} - \mathbf{F}(\mathbf{x}, \mathbf{b}))^T \mathbf{S}_\epsilon^{-1}(\mathbf{y} - \mathbf{F}(\mathbf{x}, \mathbf{b}))\right] + \frac{1}{2}(\mathbf{x} - \mathbf{x_a})^T \mathbf{S_a}^{-1}(\mathbf{x} - \mathbf{x_a}). \tag{1}$$

The measurement vector is represented by $\mathbf{y}$, $\mathbf{F}$ is the forward model for the lidar, $\mathbf{x}$ is the vector containing all retrieval parameters, $\mathbf{b}$ is the forward function parameter vector, $\mathbf{S_a}$ is the covariance matrix of the *a priori* values, and $\mathbf{S_\epsilon}$ is the measurement covariance matrix. The cost function is consists of two terms. The first term is the weighted least-squares regression. The second is a regularization term, which provides additional information to the solution of an *a priori* state. The *a priori* covariance matrix and the measurement covariance matrix define the solution space of the retrieval. Minimizing the cost function produces the retrieval solution ($\hat{x}$), where the solution is then the maximum *a posteriori* solution based on the probability distribution functions and is given by:

$$\hat{\mathbf{x}} = \mathbf{x_a} + (\mathbf{K}^T \mathbf{S_\epsilon}^{-1} \mathbf{K} + \mathbf{S_a}^{-1})^{-1} \mathbf{K}^T \mathbf{S_\epsilon}^{-1} (\mathbf{y} - \mathbf{F}(\mathbf{x_a})) = \mathbf{x_a} + \mathbf{G}(\mathbf{y} - \mathbf{F}(\mathbf{x_a})), \tag{2}$$

where $\mathbf{K}$ refers to the Jacobian matrix, $\mathbf{G}$ is the gain matrix. The Jacobian matris is defined as $\frac{d\mathbf{y}}{d\mathbf{x}}$ and the Gain matrix is $\frac{d\mathbf{x}}{d\mathbf{y}}$. An in depth description of OEM theory applied to atmospheric physics can be found in Rodgers (2000). Sica and Haefele (2016) discussed the initial application of OEM to Raman lidar water vapour retrievals. Details regarding the changes to the original water vapour retrieval code can be found in Appendix 4B of Hicks-Jalali (2019). Other lidar OEM retrievals and their applications are discussed in Sica and Haefele (2015) and Jalali et al. (2018) for Rayleigh temperature profile retrievals, Mahagammulla Gamage et al. (2019) for Raman temperature profile retrievals, Farhani et al. (2019) for differential absorption lidar (DIAL) ozone profile retrievals, and Mahagamulla Gamage et al. (2020) for Raman relative humidity profile retrievals.

Processing the entire RALMO time series requires an ongoing calibration of the lidar. We have combined GRUAN radiosondes and implemented a solar background calibration method to allow a consistent calibration across the 11.5 year time series. The continuous calibration is an internal calibration technique using the solar background between the nitrogen and water vapour channels and was first introduced in Sherlock et al. (1999). The internal solar calibration method produces a relative calibration function which is scaled to the external calibration using the GRUAN-corrected radiosondes. The calibration function has an uncertainty of 5% of the calibration value and a corresponding uncertainty of 5% in the final water vapour mixing ratio. The uncertainty in the solar calibration method is the same as what would be introduced by using GRUAN-corrected radiosondes for external calibration (Hicks-Jalali et al., 2019). However, by using an internal calibration function, the lidar trends remain mostly independent of an external instrument. The details of this calibration are given in Appendix A. This new calibration is consistent with the GRUAN calibration of RALMO conducted in Hicks-Jalali et al. (2019).

One of the advantages of using an OEM retrieval over the traditional method (Whiteman et al., 1992; Whiteman, 2003) is the addition of the uncertainty budget and averaging kernels for each profile. The addition of the averaging kernels in particular is important because they may be used to more accurately compare results with other instruments which utilize OEM retrievals such as satellite-based limb-sounding instruments, fourier transform infrared spectrometers, or microwave radiometers. Another advantage of using OEM for lidar measurement analysis is that the measurements do not need to be corrected before being used for the retrievals. It can be more difficult to accurately propagate uncertainties through corrections to measurements which would prevent a complete uncertainty budget from being produced on a profile-by-profile basis. Whiteman et al. (2012) provided robust estimates of a total uncertainty budget associated with the traditional water vapour method, however, the exact values would depend on each individual system. Leblanc et al. (2016) suggests a standardized method of calculating uncer-

tainty budgets for the NDACC group, which is rigorous, but difficult to implement on a profile-by-profile basis. Additionally, corrections to the raw measurements can further induce uncertainties in the final product which may not be accounted for. Typical corrections for water vapour measurements include: accounting for photomultiplier paralysis (dead time), background noise, overlap, differential aerosol transmission, and sometimes merging multi-channel measurements. The last of these can

result in unknown uncertainties and biases in the water vapour, and is not necessary in OEM since the final retrieval is one profile which has been retrieved using all available measurements (Sica and Haefele, 2016).

While the OEM has its advantages, some disadvantages of the method are that it is more computationally intensive than the traditional ratio method and that it is more difficult to implement. Nevertheless, a single water vapour retrieval does not take more than 30 seconds to run on an average personal laptop and the method is still quite practical for automatic and consistent

processing of large data sets such as RALMO's. Longer run times occur when more variables are retrieved or when the bin size of the retrieved profiles are small.

We do not correct the RALMO measurements for the aforementioned possible signal effects; however, we have done some minor pre-processing before the measurements are entered into the OEM retrieval. We used nightly-integrated profiles in order to maximize our altitude coverage in the troposphere. It is not possible to simply integrate (sum) all profiles in one night

due to the possible large variability in signal strength caused by clouds passing in and out of the lidar's field-of-view (FOV). Clouds can completely attenuate the nitrogen and water vapour signals depending on their composition and optical thickness. To ensure that the signal throughout the night is free of optically thick clouds, we applied a cloud mask to all raw profiles between the start of astronomical twilight after sunset until the end of astronomical twilight the next morning. The cloud filter required the nitrogen signal-to-noise ratio to be at least 1 at 10 km altitude and also the background be no higher than 10 photon

counts/bin/min. We found that these criteria effectively removed scans measured in the presence of optically thick clouds, and only left scans measured in the presence of optically thin/semi-transparent clouds such as cirrus clouds. Cirrus clouds and other aerosol layers are accounted for through the aerosol extinction retrieval in the OEM algorithm (Sica and Haefele, 2016; Hicks-Jalali, 2019).

The raw nitrogen and water vapour profiles that remained after the cloud mask was implemented were summed to produce

one profile for each digital and analog channel for a total of 4 profiles per night. Due to the fact that the effect of clouds will vary nightly, the measurements for each night do not contain the same number of raw profiles and thus, have varying integration times. We required that all retrieved nightly water vapour profiles have at least 30 min of measurements over the course of a night. The maximum integration time was 10 hours on a clear dry night during the winter. The minimum of 30 min was chosen to make sure we included low-signal nights in the climatology and did not bias the results towards cloud-free conditions. Thirty

minutes is also the amount of time that a radiosonde typically needs to reach the tropopause at mid-latitudes. However, we do recognize that the lidar measurements in this study are naturally biased towards high pressure system conditions, since current Raman water vapour lidars require clear to semi-clear conditions in which to operate. We will compare the nighttime only PWV water vapour trends to daytime radiosonde trends in Sect. 4.2 and discuss the daytime/nighttime bias.

After applying the cloud mask and creating the temporally summed profile, the resulting profiles at 3.75 m resolution were

again summed in altitude to produce 30 m altitude bins. The final input to the OEM algorithm is a single "nightly-integrated"

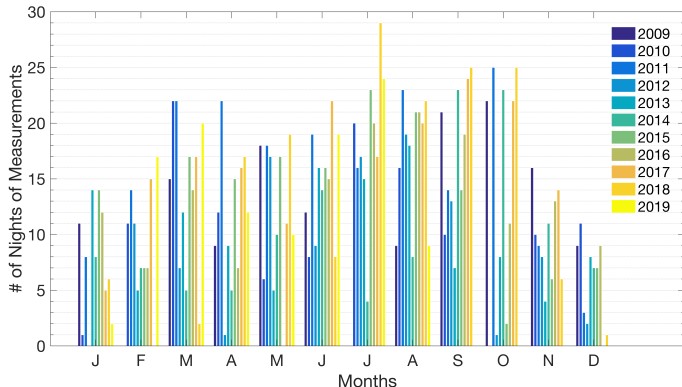

**Figure 1.** The total number of semi-clear nights with at least 30 minutes of profiles per month from 2009 through the middle of August 2019. Winter months have fewer measurements due to more cloud cover and precipitation. Most measurements are taken in the summer.

profile with an altitude bin size of 30 m. The altitude bin size of the OEM retrieval grid is 90 m, or the minimum resolution which allows the profiles to reach near the tropopause. The total distribution of lidar measurements over each year is shown in Fig. 1. The histogram shows the number of retrieved profiles per month for each year of measurements and is the maximum number of measurements available. The largest number of measurements occur in the summer, when there are more favorable

meteorological conditions; however, the nights are shorter than in winter. The least number of measurements occur in December and January when there are less favorable conditions, such as a higher frequency of low clouds and precipitation. December and January are also missing a few years of measurements due to the replacement of the laser in late 2017 and early 2018. December of 2018 was extremely cloudy and there were no clear or semi-clear nights available for water vapour measurements. However, most months have at least 9 years of measurements.

It is important to note that the lidar measurements in this study are limited to only nighttime during periods of favourable weather. This selection introduces a bias in the lidar results, which we will discuss later. Despite the nighttime bias, Wang et al. (2016) noted that nighttime measurements are particularly important for understanding water vapour feedback contributions, as nighttime precipitable water vapour trends are better correlated with surface temperatures. The higher correlation at night is likely due to fewer shortwave radiation sources such as clouds, aerosols, surface evaporation, surface albedo, and a higher

contribution from long wave radiation which is the dominant factor in the water vapour greenhouse effect (Dai et al., 1999; Wang et al., 2016).

## 2.2 Radiosondes

### 2.2.1 GRUAN Radiosondes

GCOS Reference Upper Air Network (GRUAN) certified radiosondes are currently the highest quality radiosonde data product

available. These radiosondes have been well characterized and corrected for several biases (Dirksen et al., 2014). Unique to

GRUAN radiosonde products is the calculation of measurement uncertainties as a function of altitude which allows for better comparison between radiosondes and other instruments. The aerological station of MeteoSwiss in Payerne, Switzerland, has launched Vaisala RS92 sondes biweekly at noon and midnight UTC since October 2011 to obtain GRUAN-certified profiles of temperature and humidity. Launches are co-located with RALMO. Unfortunately, not all of the radiosondes launched at

the station from 2009–2018 were GRUAN-compliant or GRUAN-corrected, therefore, we were unable to calculate a GRUAN radiosonde climatology or use them for trends. GRUAN-certified sondes were used to calculate the lidar calibration constants in Appendix A, as well as examine the uncertainty in the daily radiosondes (Hicks-Jalali et al., 2019).

### 2.2.2   Daily Radiosondes

Meteoswiss also launches an operational radiosonde every day at 11h00 and 23h00 UTC. The Meteoswiss operational ra-
diosonde time series uses multiple radiosonde types: the SRS400 (Martin et al., 2006; Morland et al., 2009; Brocard et al., 2013b) from 2009–2010, SRS-C34 from 2010–Jan 2017, SRS-C50 from Feb 2017–March 2018 , and the Vaisala RS41 (Jensen et al., 2016) from March 2018 - present day. The C34 and C50 radiosondes are manufactured by MeteoLabor. The C34 uses a Sippican hygristor for humidity measurements which is quoted to have an accuracy of 2% RH (Meteolabor, 2010); however, no public validation studies have been published. The C50 is the updated version of the C34, however, no public documenta-
tion exists on the C50 specifications. As these radiosondes are not GRUAN-certified we do not use them to compare with the individual lidar profiles. However, we have compared their PWV measurements to that of the coincident GRUAN-processed RS92 radiosondes. We compared the operational sonde PWV measurements to coincident GRUAN PWV measurements and found an average difference of $0.8\% \pm 3.7\%$ PWV between the two sets of sondes. The scatter in the radiosonde measurements did not change between the different types of radiosondes, therefore we have not attempted any homogenizing between the
different data sets. The average difference between the two sets of sondes was within the GRUAN uncertainty for PWV, there-
fore, we were comfortable using the operational sondes for PWV comparison with RALMO. We have also used the pressure measurements from these radiosondes to convert the retrieved water vapour lidar altitude profiles onto a standard pressure grid.

## 3   A Monthly Tropospheric Water Vapour Climatology for Switzerland

Climatologies are extremely useful as they provide a baseline for daily measurement comparison and can be used as *a priori*
information for forecast modeling and reanalysis models. To that end, we have calculated a monthly water vapour climatology using the RALMO measurements for Payerne, Switzerland. A monthly climatology was chosen instead of a daily climatology due to the inability to retrieve enough coincident daily measurements to accurately represent a daily average.

A Raman lidar's native measurement grid is in altitude, however, models and satellites often work on pressure grids. There-
fore, we have interpolated our final retrieved profiles onto a pressure grid to facilitate comparisons with other studies. The
climatology data as a function of altitude and pressure has been made available via the Zenodo database (Hicks-Jalali et al., 2020). We interpolated each profile onto a standard pressure grid from 950 hPa to 100 hPa using pressure and altitude data from

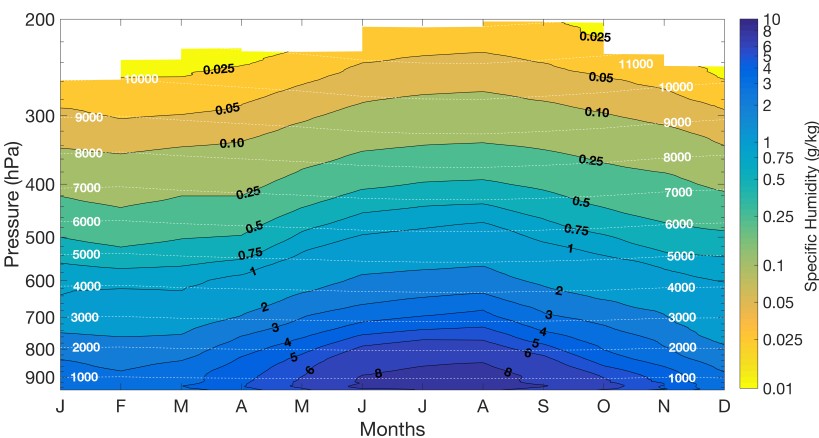

**Figure 2.** The monthly climatology for tropospheric water vapour from RALMO in units of specific humidity (g/kg). Contours for specific humidity are labeled in bold black. White dashed lines are the average altitudes for each pressure level and are in units of meters. Larger amounts of water vapour are seen in the summer months due to higher temperatures. The highest altitude/lowest pressure is 250 hPa during the summer months. In the winter, the lowest pressure is 280 hPa due to the lower water vapour content. Higher water vapour content is shown in darker blue colors and lower water vapour content in lighter colors.

the operational radiosondes launched at midnight UTC. When surface pressures are higher than 950 hPa, the measurements are extrapolated. No surface pressure measurements were higher than 960 hPa.

Once the lidar profiles have been interpolated, all profiles in a month which pass the cost threshold of 3.5 are averaged together and weighted based on their statistical uncertainties. Due to the cloud masking applied to each individual profile, it is possible that some altitude bins will use fewer measurements than others. We required that each individual bin have measurements from at least 5 different years, with at least 3 measurements per year to calculate a representative average state. This means that the minimum number of measurements is 15 per pressure bin. Figure 2 shows the resulting specific humidity climatology with respect to pressure. The corresponding average altitudes for each pressure are shown as white contour lines.

The climatology shows the expected seasonal cycle for water vapour with high concentrations (darker blues) in the summer and lower concentrations in the winter. The spring months are slightly drier than the fall months, with specific humidity values only starting to increase in April and high summer concentrations lasting through September. On average, the lidar is able to retrieve measurements consistently (at least 15 profiles per bin over 5 years) up to 280 hPa (roughly 10 km). However, in the summer months, the climatology reaches up to 250 hPa (roughly 12 km) due to higher lidar SNRs from the higher water vapour content.

The average statistical uncertainties for the climatology are consistent across all seasons, with the exception that the summer months and September show smaller uncertainties at lower altitudes due to their larger SNRs (Fig. 3). The average statistical uncertainties are calculated by taking the average of the uncertainties for all the profiles used in each month. The statistical uncertainties are presented as the uncertainty in the water vapour mixing ratio, and not the uncertainty in specific humidity. The

OEM retrieval is in units of water vapour mixing ratio, and while we have converted the final retrieval profiles to units of specific humidity, even at the highest concentrations of water vapour the difference between the two units is less than 2%. Therefore, the uncertainties calculated for the water vapour mixing ratio are comparable to the uncertainties for the converted specific humidity profile. The strength of the Raman lidar signal and the statistical uncertainty is dependent on the amount of water

vapour present in the atmosphere. Therefore, high statistical uncertainties are associated with low specific humidity levels. At high pressures where more water vapour is present, such as in the boundary layer, the statistical uncertainty is less than 1%. However, at lower pressures and near the tropopause where water vapour quantities are low, the statistical uncertainties reach an average of 14%. In the winter, when the air is drier, we see slightly higher uncertainties than in the summer at the same altitudes. At the surface, 82% of the profiles used in this study have statistical uncertainties of less than 5% and 96% of the

profiles have statistical uncertainties less than 10%. At 250 hPa, 77% of the profiles have uncertainties lower than 20%, while the remaining 23% had uncertainties between 20 and 25%.

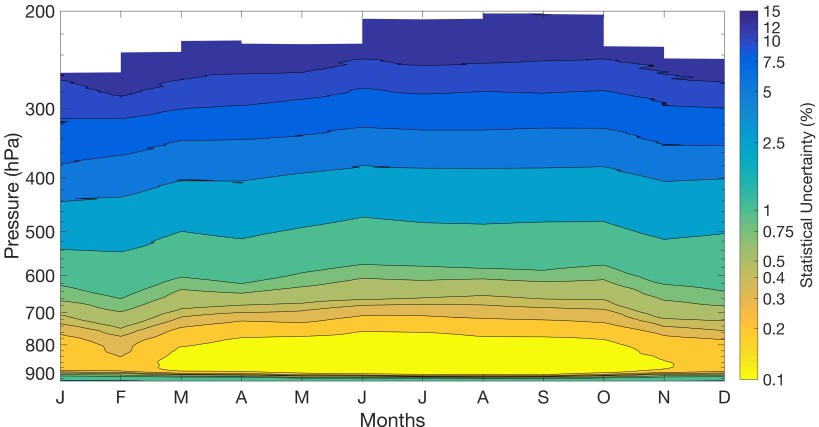

**Figure 3.** The average statistical uncertainties (represented as percent uncertainties) for each month as a function of pressure. The highest statistical uncertainty is 14% of the water vapour mixing ratio between 275–250 hPa or 11–12 km.

In this study we refer to the individual retrieval uncertainties in water vapour mixing ratio due to uncertainties in non-retrieved forward model parameters as parameter uncertainties (Fig. 4). Parameter uncertainties are calculated via Gaussian uncertainty propagation for each retrieval parameter. Figure 4 shows the parameter uncertainties' effect on water vapour mixing

ratio calculated for each retrieval, including: the calibration constant, Ångstrom exponent, NCEP air density, aerosol extinction profile, overlap, and the Rayleigh cross-section. Details regarding the covariances of the system parameters can be found in Appendix 4B of Hicks-Jalali (2019). Every uncertainty profile used in the climatology is shown in gray dashed lines and the average uncertainty for each component is the black dashed line. Note that systematic uncertainties are on a different height scale for the aerosol extinction profile and overlap components. This choice is because these forward model parameters are

also retrieval parameters for part of the profile. Overlap is retrieved from the ground to 400 hPa, therefore the uncertainty in overlap on the retrieved water vapour mixing ratio is only calculated above 400 hPa. The opposite situation occurs for the

retrieved aerosol extinction; it is retrieved above 400 hPa and therefore its uncertainty on the water vapour mixing ratio is only calculated below 400 hPa. The largest systematic uncertainty, which dominates the retrieval, is the 5% calibration uncertainty. Sica and Haefele (2016) originally assigned a value of 5% which was later confirmed by the study conducted in Hicks-Jalali et al. (2019). The residuals of the smoothing spline fit to the combined calibration time series also showed an average of

5% variation in the residuals (Appendix A). The second largest component is the uncertainty due to the assumption of the aerosol Ångstrom exponent. The OEM retrieval assumes that the Ångstrom exponent does not change with altitude, which can cause larger uncertainties in clouds where our assumed value may not be appropriate. We used a mean climatological value of $1.5 \pm 0.5$, which results in an average uncertainty on the order of 1% in the water vapour mixing ratio. In 4 out of the 1300 nights used in this study, the uncertainties caused by the uncertainty in the Ångstrom exponent were larger than 5% due to the

Ångstrom exponent choice being a poor fit in a persistent cloud. The uncertainty due to NCEP model air density is slightly higher than the Ångstrom exponent uncertainty at altitudes below 2 km, but with an average uncertainty of around 0.25% for all altitudes. All other systematic uncertainties including extinction, overlap, and the Rayleigh cross section are less than 0.1%. However, the Rayleigh cross section uncertainties also had 4 nights which exhibited uncertainties larger than 0.5%. The major improvement to the water vapour retrieval since Sica and Haefele (2016) is the retrieval of overlap from the surface to 6 km (if

there are no clouds below 6 km). In Sica and Haefele (2016), overlap was not a retrieval parameter and added another 7 - 10% uncertainty below 3 km.

We also compared the lidar PWV measurements to the PWV column from the Payerne daily radiosondes during the same time period as the height-resolved climatology. The radiosonde measurements are systematically larger than than the lidar measurements, likely due to the fact that the lidar is not able to take measurements in the first 100 m from the surface. The lidar

PWV climatology agrees with the operational radiosonde PWV climatology within their respective uncertainties (Fig. 5). The uncertainties for the climatology were calculated using the standard deviation of all measurements in one month.

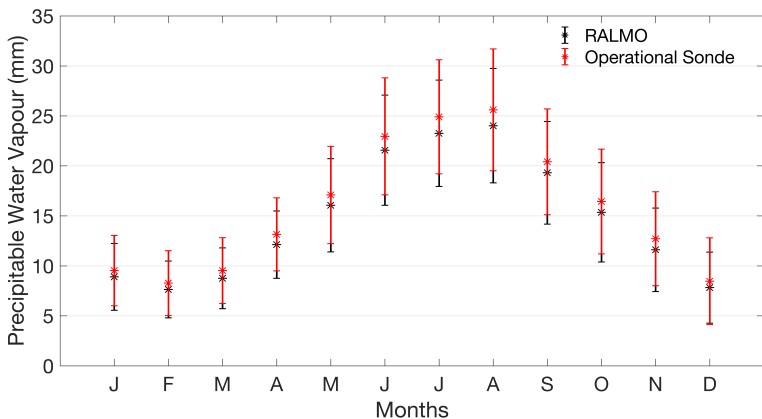

**Figure 5.** The PWV climatology of both the lidar and the operational radiosondes. The error bars are the $1\sigma$ standard deviation of all of the measurements in each month

. The climatology was calculated using the same dates for both the radiosonde and the lidar measurements.

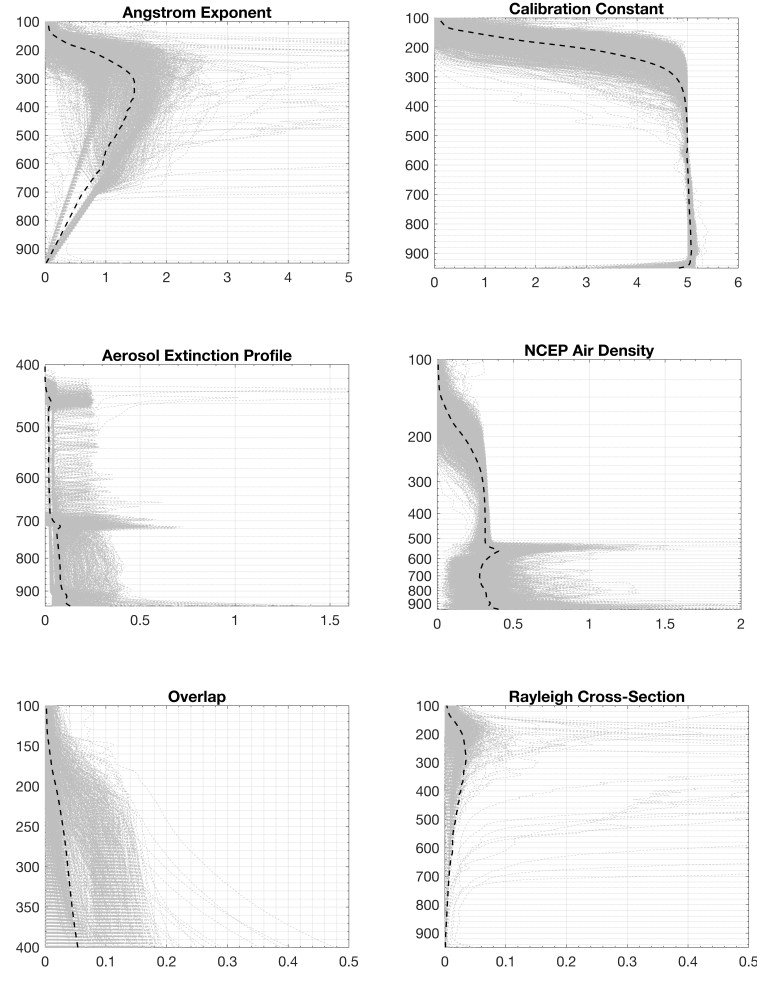

**Figure 4.** The water vapour mixing ratio systematic uncertainty budget for each night used in the water vapour climatology (gray dashed lines), and the average uncertainty contribution from each component (black dashed line). The largest contributors are the Ångstrom exponent and the calibration constant. Four nights out of 1300 had Ångstrom exponent uncertainties larger than 5% due to our assumed value not fitting with persistent clouds. The average uncertainty contribution from the NCEP air density is .25%. All other uncertainties contribute less than 0.1% on average.

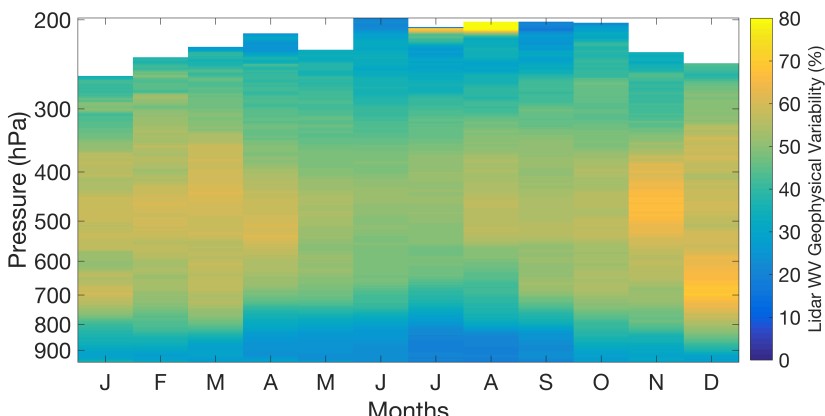

**Figure 6.** The monthly geophysical variability of water vapour measured by RALMO as a function of pressure. The largest amount of variability can be seen in the free troposphere between 600 and 400 hPa in the summer, and from 800 to 450 hPa in the winter.

The PWV climatology shows the same behavior as the profile climatology already shown in Fig. 2, with the largest amount of water vapour in the summer months and smallest amounts in the winter months.

### 3.1 Geophysical Variability

Water vapour concentrations change by up to four orders of magnitude through the troposphere. The troposphere is also a region of many dynamic processes, including the hydrologic cycle as well as general global circulation. These processes have a direct impact on water vapour variability and make it difficult to measure. The abundance of measurements used in this study make it possible to estimate water vapour's variability as a function of altitude. The geophysical variability ($\sigma_{geo}(z)$) of the specific humidity can be estimated by taking the standard deviation of all profiles in one month ($\sigma_w(z)$) and removing the measurement variability, which is dominated by the statistical uncertainty ($\sigma_{stat}(z)$, Eq. (3), Argall and Sica (2007)). The remaining variability should then be physical in nature.

$$\sigma_{geo}^2(z) = 100 \frac{(\sigma_w^2(z) - \sigma_{stat}^2(z))}{\langle q(z) \rangle^2} \tag{3}$$

The percent geophysical variability for water vapour was calculated using Eq. (3) in which $\sigma_w^2(z)$ is the variance of all monthly profiles, $\sigma_{stat}^2(z)$ is the average of the statistical uncertainty variances of the retrievals, and $\langle q(z) \rangle$ is the climatology. Figure 6 is then the geophysical variability in units of percent variability from the mean state.

The smallest percent variability is in the boundary layer, typically in the summer. However, as this is a nighttime and primarily clear-weather climatology, the boundary layer should be stratified and either neutral or stable at these times. Therefore, Fig. 6 would represent the variability of the boundary layer during nighttime and primarily high-pressure conditions.

Above the boundary layer, from 700 hPa in the winter/spring/fall and up to 350 hPa there is much higher variability present. The largest variabilities of 60% or higher are located between 550 - 400 hPa during the spring and fall, and between 750 -

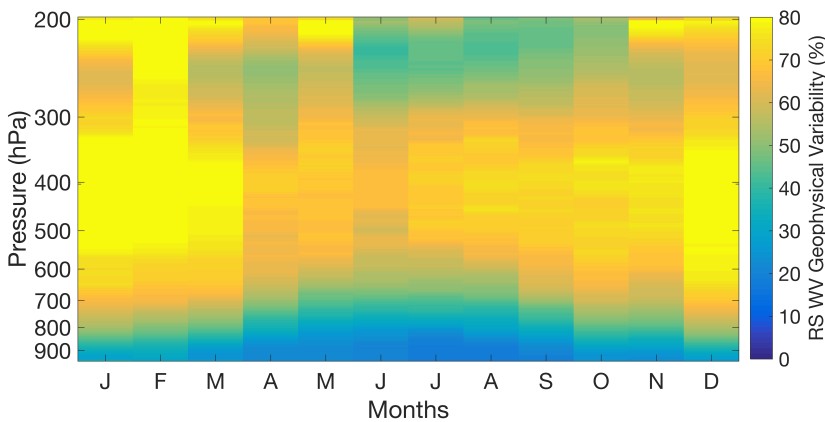

**Figure 7.** The water vapour geophysical variability estimated using all available daytime and nighttime operational radiosondes as a percentage and function of pressure. The radiosondes see the same behavior as the lidar, and the distribution of the variability is the same, although the magnitude is generally 20% larger. The larger magnitude is possibly due to the underestimation of the daily radiosonde uncertainties.

600 hPa in December. The latter region in December is likely due to sampling bias since December had half the measurements as the summer months. However, the high variability regions between 600 hPa and 400 hPa are interesting and could represent dynamic processes at work. The region from 600 to 400 hPa is a region of active mixing of air from the upper troposphere or stratosphere and the surface. Simultaneously, planetary waves transport air horizontally from different air masses at these
heights. Therefore, we would expect the region from 600 hPa to 400 hPa to exhibit a high amount of variability.

We compared the lidar-measured geophysical variability to the water vapour geophysical variability measured by the operational radiosondes. All available daytime and nighttime operational radiosondes were used to calculate the geophysical variability. The total value of the percent geophysical variability is only an estimate because we do not know the uncertainty of the daily radiosonde's humidity measurements. The maximum uncertainty reported by the GRUAN radiosondes in the tropo-
sphere is around 10% of the mixing ratio measured. Therefore, we assume a constant uncertainty in the operational radiosondes measurements of 10%, although this would be a lower limit of the uncertainty of the routine sondes because they have not been corrected. The variability is then calculated in the same way as for the lidar (Fig. 7).

The radiosondes measure similar behavior in the variability to the lidar, however, the magnitude of the variability is roughly 20% larger than the lidar's. As discussed earlier, half of this difference is likely due to underestimating the uncertainty in
the operational radiosondes. However, it could partially be due to the fact that more measurements are available from the radiosonde than from the lidar. When comparing November and December profiles, it can be seen that the regions of large variability in November and December in the lidar variability are likely due to sampling bias and not physical. However, it is encouraging to note that the same physical structure is seen in both instruments. This result would suggest that the large variability in the troposphere is not due to clouds or layers affecting the climatology, but likely due to the increased mixing
of air between 600 to 400 hPa in the summer, and 700 to 450 hPa in the winter. In high pressure systems, the air from the

stratosphere and high regions of the troposphere converge which causes air to subside at the surface (Holton, 2004). The colder air in the stratosphere and lower troposphere is usually drier. This exchange of moist air and dry air in the free troposphere could explain the large variability. It is interesting to see that adding the daytime sondes to the variability calculation does not change the overall behaviour of the water vapour in the troposphere. We explored looking at the difference between the variability including daytime sondes and not including the sondes and found almost no difference in the variability when the daytime sondes were added. The radiosondes and the lidar measure more variability in the winter months than in the summer months. European winter weather is highly influenced by the polar jetstream. An increase in the variability of water vapour during these months could be caused by an increase in the exchange of air between the Arctic and the mid-latitudes.

## 4 Deriving Trends

There have been several tropospheric water vapour trend studies conducted over the course of the last few decades, only a few of which are cited here (Hense et al., 1988; Ross and Elliott, 1996; Ross and Elliot, 2001; Morland et al., 2009). Whiteman et al. (2011a) investigated the potential for water vapour Raman lidar trend calculations and carefully estimated the thresholds needed to calculate statistically significant trends in the upper troposphere. They found that with no statistical uncertainty, one would need between 10 to 12 years of daily measurements to calculate trends at 200 hPa due to water vapour's naturally large variability at that pressure. We have 11.5 years of measurements and are therefore in the realm of possible detection at the 95% level; however, whether or not we can detect trends depends on the variability of the atmosphere at the levels we choose and the magnitude of the trend itself. As we will show, the variability of the atmosphere in the free troposphere will determine our ability to detect trends at the 95% level. According to Weatherhead et al. (1998), trends of at least 5% per decade should be calculable in 10 years provided the standard deviation of the noise in the trend residuals is less than 6% and the autocorrelation of the trend fit residuals is less than 0.4 (Table 1 of Weatherhead et al. (1998)). Therefore, the lowest pressure we should be able to calculate trends of at least 5% is around 350 hPa (11 km) provided the autocorrelation is small and we have relatively little noise in our trend residuals.

### 4.1 Trend Estimate Methodology

The bootstrap method of Gardiner et al. (2008) makes use of a least-squares regression for a model which includes both the linear trend and the annual and semi-annual oscillations via a Fourier function (Eq. 4):

$$
\begin{aligned}
f(t, a, \mathbf{b}) = at + b_0 &+ b_1 \cos(2\pi t) + b_2 \sin(2\pi t) \\
&+ b_3 \cos(4\pi t) + b_4 \sin(4\pi t) \\
&+ b_5 \cos(6\pi t) + b_6 \sin(6\pi t),
\end{aligned}
\tag{4}
$$

where $a$ is the trend in units of either g/kg per year or mm/year, the vector $\mathbf{b}$ is comprised of all the seasonal variation coefficients ($\mathbf{b_n}, \mathbf{n} = \mathbf{0, 1, ..6}$) as well as the offset ($\mathbf{b_0}$), and $t$ is time. Gardiner et al. (2008) found that, for their study, it was

not necessary to continue beyond the third order of the Fourier series, as doing so no longer decreased the RMS of the residuals. We also found no difference in the residual RMS by adding terms beyond the third order in the Fourier series. Indeed, the third order could probably have been removed as well, however, we have kept it to maintain consistency with other studies.

We calculated trends using monthly averages of nightly averaged measurements from RALMO as well as the nightly radiosondes. Monthly averages were used to reduce the autocorrelation of the measurements and increase the statistical significance of the trends. The statistical significance of the trends was determined by using the uncertainty equation from Weatherhead et al. (1998) which includes the autocorrelation coefficient of the residuals:

$$\sigma_a = \frac{\sigma_N}{n^{3/2}} \sqrt{\frac{1+\phi}{1-\phi}}. \tag{5}$$

The uncertainty of the trend, $\sigma_a$, is determined from $\sigma_N$, the standard deviation of the residuals from the least squares fit, $n$, the number of years of measurements (in our case 11.5), and $\phi$, the autocorrelation coefficient at lag 1. All uncertainties presented in this study are $2\sigma_a$ to determine if the corresponding trend is statistically significant at the 95% level.

As shown in Fig. 1, some months of data are missing from the full time series. However, there were no successive gaps in the monthly-averaged time series. We linearly interpolated the monthly-average time series before calculating the trend to fill the 6 gaps in the missing months and to calculate meaningful autocorrelation coefficients.

## 4.2 Precipitable Water Vapour Trends

We first calculated tropospheric precipitable water vapour (PWV) trends using RALMO and the daily radiosonde measurements to compare with the majority of previous studies where height-resolved measurements were not available. Precipitable water vapour values were calculated by integrating water vapour densities between the daily station surface pressure as measured by the daily radiosonde and 280 hPa to maintain the same pressure range over the entire year. Table 1 shows the value of the trends for the 30-day averaged RALMO and daily radiosonde nighttime measurements from January 2009 to August 2019. The PWV measurements, the function fits, and the trend lines are shown in Fig. 8. We calculated four radiosonde trends to compare with the RALMO PWV trend. The first two methods used only nighttime data, but the first limited the data to the same nights as the radiosonde and the second used all available nighttime radiosondes. They are shown in rows 1 and 2 of Table 1, respectively. The difference between the two nighttime radiosonde trend methods could represent the bias from using only semi-clear nights during clement weather. However, the bias between the two radiosonde nighttime trends is more likely caused by the number of points used in each trend and the uncertainty in the radiosonde measurements. Roughly 25% of the possible nights are used in the trend analysis for the coincident lidar dates and as the radiosondes do have a larger and uncharacterized uncertainty it is more likely that the large trend value is due to a larger scatter in the radiosonde measurements. Given that the lidar trend is much closer to the radiosonde trends which use all available nights, including the radiosonde trends calculated using daytime and nighttime measurements, the large radiosonde trend using only nights consecutive with the lidar is more likely due to random error and not a difference in weather. We also calculated daytime radiosonde trends using all available daytime measurements as well as a combined trend using all available daytime and nighttime measurements (rows 3 and 4 of Table 1). The daytime

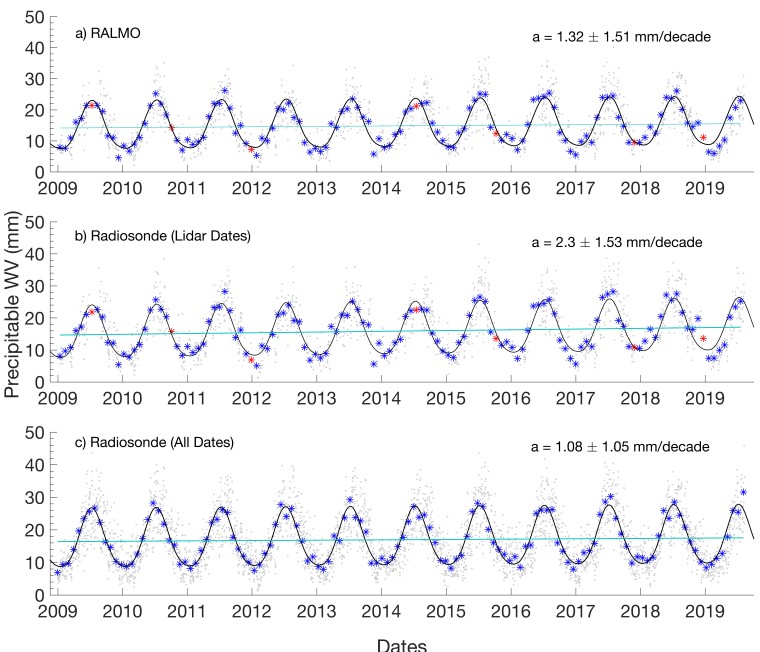

**Figure 8.** The nighttime precipitable water vapour measurements taken by RALMO and the daily radiosondes. For all subplots, the nightly measurements are the grey dots, 30-day averages are the blue stars with interpolated points in red, the fitted function is in black, and the trend line is in light blue. a) Ralmo PWV measurements, b) Radiosonde PWV measurements on coincident dates with lidar measurements, c) Radiosonde PWV measurements for all available dates.

**Table 1.** Table of precipitable water vapour trends calculated using RALMO and daily radiosonde monthly averaged measurements from January 2009 - August 2019. Trends are presented in units of mm per decade (mm/dec), percent change from the mean per decade (%/decade), and percent change in water vapour per °C. The trend values in column 3 were calculated by dividing the water vapour trend in column 2 by the surface temperature trend of $1.38 \pm 1.41$ °C per decade. Trend uncertainties are $2\sigma_a$ values and in in units of mm per decade and are calculated using Eq. (5).

| Measurement Series | Trend (mm/dec) | Trend (%/dec) | Trend (%/°C) | Trend Uncertainty (mm/dec) | $\phi$ | Significance Level |
|---|---|---|---|---|---|---|
| RALMO | 1.32 | 8.85 | 6.41 | 1.51 | 0.21 | 90% |
| Daily Radiosonde (Lidar dates) | 2.31 | 14.45 | 10.59 | 1.53 | 0.16 | 95% |
| Daily Radiosonde (All dates, night only) | 1.08 | 6.36 | 4.94 | 1.05 | 0.09 | 95% |
| Daily Radiosonde (All dates, day only) | 1.18 | 7.20 | 5.27 | 0.97 | 0.05 | 95% |
| Daily Radiosonde (All dates, day and night) | 1.19 | 7.20 | 5.27 | 0.98 | 0.07 | 95% |

radiosonde trends differ from the nighttime radiosonde trend using all available nights by 0.1 mm/decade and they agree within their 1-sigma uncertainties.

The lidar and radiosonde precipitable water vapour trends have varying levels of significance. The RALMO trend is significant above 90% while the radiosonde trends are significant above 95% (Table 1). The trend calculated using all available nighttime radiosonde measurements is slightly smaller than the RALMO PWV trend, but slightly larger than previously calculated trends for Payerne and the region (Morland et al., 2009; Wang et al., 2016; Nyeki et al., 2019).

The Clausius Clapeyron relationship suggests that the water vapour in the atmosphere should increase by roughly 7.5% per $1 °C$, or more depending on the initial temperature (Held and Soden, 2000). Therefore, such large PWV changes should be correlated with proportionally large increases in temperature. We used the temperature measurements at 500 m (950 hPa) a.s.l. from the daily radiosonde to compare with the lidar water vapour trend because the lowest available measurements are at 550 m. We assume that the lidar PWV measurements should correspond best with the temperatures at that altitude. The same nights used in the lidar and radiosonde water vapour trend analysis were used for the radiosonde temperature measurements. The 950 hPa temperature trend was statistically significant at 95% with a slope of $1.38 \pm 1.41 °C$ per decade. The temperature trend we measure is larger than that measured by Morland et al. (2009) and Nyeki et al. (2019). However, the percent change in water vapour per degree for all PWV trend values is between 5 - 10%, which is consistent with the expected change assuming relative humidity is conserved. The differences between our temperature trend measurements and other studies will be discussed further in Sect. 5.

## 4.3 Height-Resolved Trends

Here we show height-resolved water vapour trends measured at 11 layers of the troposphere from 950 hPa to 250 hPa. Each layer is an average of measurements from $\pm 10$ hPa from the center (e.g. measurements at 900 hPa are an average between 890 - 910 hPa). The layers were averaged to reduce the variability due to choosing a single pressure level. We calculated the trend for each layer using monthly averages of the time series. We calculated the specific humidity height-resolved trends using the same bootstrapping method that was used for the PWV trends.

An example of the trend at 800 hPa is shown in Fig. 9. The blue asterisks are the monthly averages, with the interpolated points shown in red. The nightly points are shown in light gray. The trend and seasonal fit are in light blue and black, respectively.

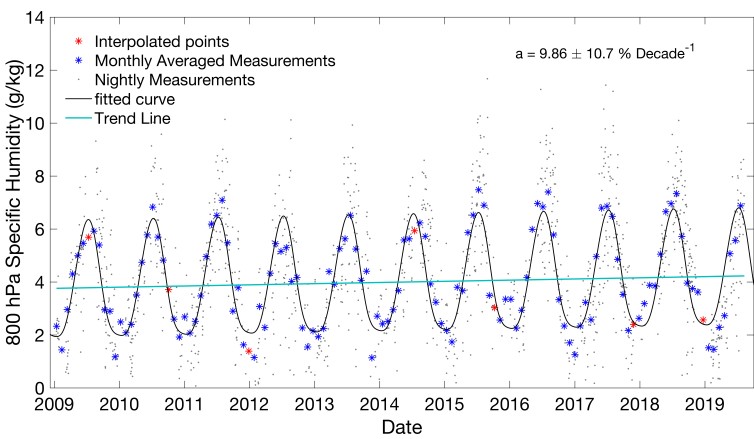

**Figure 9.** An example lidar specific humidity trend at a fixed pressure level of 800 hPa showing the seasonal fit (black) and the linear trend line (light blue). Nightly measurements are in gray, monthly averages are blue asterisks with the 7 interpolated monthly points in red. The trend is calculated over the series which includes the interpolated points.

The trends at each layer are shown in Table 2. The trend at 950 hPa is statistically significant at the 95% level, while trends at 800, 700, 600, and 350 are significant at the 90% level. The trend at 400 hPa is statistically significant at the 80% level. The rest of the trends have significance levels of less than 80%. The trends in Table 2 are presented in absolute units per decade, percent change from the mean per decade, as well as percent change from the mean per degree. The uncertainties are presented as $2\sigma_a$ in units of percent change from the mean per decade. All trends were positive, but varied between 3 and 16% per decade depending on the layer. The trend at 950 hPa is statistically significant, but the noise in the residuals is less than the noise at 900 hPa.

To calculate the change in the specific humidity per degree, we also calculated layered temperature trends using the daily radiosondes. The radiosonde temperature measurements were also averaged over the same pressure ranges as the specific humidity trends and averaged monthly. The temperature trends are presented in Table 3. All available nights were used to calculate the temperature trends. We found all temperature trends to be positive, with the largest 1.38°C/decade at 950 hPa. All of the temperature trends were statistically signficant at the 95% level. The temperature trends are larger by about 0.4 - 0.8 °C/decade than what was reported in Morland et al. (2009) and Brocard et al. (2013a). Interestingly, dividing the height-resolved water vapour trends by the temperature trends does not produce the expected 7%/°C relationship which was seen in the precipitable water vapour trends, with the exception of the trend at 950, 300, and 250 hPa. Instead, we see increases on the order of 10 - 17%/°C, between 800 and 300 hPa. The rest of the layers present humidity changes with temperature that would indicate that relative humidity may not be conserved at the individual layers. The PWV trends are likely dominated by the surface water vapour trend, hence why it PWV and 950 hPa trends are similar.

**Table 2.** Table of RALMO specific humidity trend calculations for each pressure layer. The first column is the specific humidity trend in units of g/kg per decade, the second column is the specific humidity trend in percent per decade, the third column is the specific humidity trend in units of percent change per degree. The fourth and fifth columns are the uncertainties of the trend in units of percent per decade, and the autocorrelation value at lag 1. Trends marked with a *, **, and *** are statistically significant at the 80%, 90%, and 95% level, respectively.

| Pressure (hPa) | Trend (g/kg Decade$^{-1}$) | Trend (%/Decade) | $2\sigma_a$ (%/Decade) | $\phi$ |
|---|---|---|---|---|
| 950*** | 0.74 | 12.11 | 10.25 | 0.25 |
| 900 | 0.05 | 0.84 | 8.15 | 0.23 |
| 800** | 0.44 | 9.83 | 10.45 | 0.22 |
| 700** | 0.35 | 13.94 | 14.36 | 0.18 |
| 600** | 0.22 | 15.85 | 16.92 | 0.19 |
| 500 | 0.03 | 4.80 | 16.19 | 0.26 |
| 400* | 0.03 | 10.87 | 15.79 | 0.23 |
| 350** | 0.02 | 12.59 | 14.07 | 0.25 |
| 300 | 0.007 | 7.08 | 14.01 | 0.23 |
| 275 | 0.002 | 3.19 | 11.6 | 0.22 |
| 250 | 0.002 | 3.55 | 11.48 | 0.25 |

**Table 3.** Table of temperature trend calculations for each pressure layer using the operational radiosondes. The second column is the temperature trend in units of degrees per decade, the third column is the $2\sigma_a$ trend uncertainty in units of degrees per decade, the fourth column is the autocorrelation coefficient at lag 1, and the fifth column is the percent change in specific humidity per degree C. Trends marked with a *, **, and *** are statistically significant at 80, 90, and 95% level respectively.

| Pressure (hPa) | Trend (°C Decade$^{-1}$) | $2\sigma_a$ (°C Decade$^{-1}$) | $\phi$ | Trend (%q °C$^{-1}$) |
|---|---|---|---|---|
| 950*** | 1.38 | 1.02 | 0.14 | 8.80 |
| 900*** | 1.20 | 1.02 | 0.24 | 0.68 |
| 800*** | 1.05 | 1.02 | 0.21 | 9.39 |
| 700*** | 0.99 | 0.91 | 0.19 | 14.09 |
| 600*** | 0.90 | 0.86 | 0.19 | 17.64 |
| 500*** | 0.91 | 0.84 | 0.15 | 5.33 |
| 400*** | 0.95 | 0.84 | 0.15 | 11.44 |
| 350*** | 1.06 | 0.79 | 0.14 | 11.88 |
| 300*** | 1.06 | 0.61 | 0.13 | 6.67 |
| 275*** | 0.95 | 0.52 | 0.13 | 3.34 |
| 250*** | 0.52 | 0.55 | 0.17 | 7.10 |

## 5  Discussion

We have calculated a lidar climatology of the water vapour distribution above Payerne, Switzerland using measurements from 2009 to 2019 from the RALMO lidar, as well as a precipitable water vapour climatology. RALMO is one of a few lidars which has produced a published high vertical resolution water vapour climatology of the troposphere with 11.5 years of consistent measurements. , the climatology in Hadad et al. (2018) (H2018) is the only other ground-based Raman lidar published climatology (410 m, 45.77°N, 2.96°E). The climatology in H2018 shows a wet bias of roughly 2 g/kg at the surface in September and October compared to RALMO. This difference is likely due to the fact that they only have 6 years of measurements from 2010–2016. The remaining months show good agreement between the RALMO and H2018 lidar climatologies. Our RALMO climatology agrees very well with the H2018 Atmospheric Infra-Red Sounder (AIRS) on the Earth Observing System (EOS) climatology which had more measurements from 2002–2017. The AIRS water vapour climatology was calculated for the 100 km radius around Cezeaux, France. The differences between the AIRS and RALMO climatologies are less than 1 g/kg.

H2018 also calculates a standard deviation for their climatology. However, it is important to note that this is not the same as the geophysical variability that we have calculated, as they do not subtract the variability due to their measurement noise. Additionally, they do not look at the relative variability, but the absolute. They show large standard deviations in the boundary layer and lower troposphere and very small standard deviations in the upper troposphere. However, dividing their standard deviation by their climatology produces variability measurements comparable to ours of around 20 to 30% in the boundary layer and between 40 to 80% in the free troposphere. H2018 states that they see more variability in the boundary layer, however, we would argue that this is because they are looking at the absolute value of the variability, and not the percentage. The free troposphere actually has larger variability, since it has much smaller concentrations of water vapour. In summary, both the H2018 and our variabilities are consistent with each other.

The RALMO climatology is a nighttime and clear-sky only climatology. The AIRS climatology in H2018 is not cloud-filtered (or at least no mention is made of any cloud-filtering), appears to use both day and night measurements, and differs from the RALMO climatology by less than 1 g/kg. The question is whether or not a nighttime climatology for Payerne can serve as a true representation of the average state. Morland et al. (2009) and Hocke et al. (2017) calculated the diurnal PWV cycle for Bern using the TROWARA instrument and GNSS measurements. Hocke et al. (2017) found that the average PWV diurnal amplitude was 2% of the monthly mean. In other words, the total water vapour content on average changes by 2% over the course of the day. The maximum change in water vapour was 4% over the course of the day in June, and the minimum was less than 1% in the winter months. While PWV cannot accurately represent the changes at individual layers, it does serve to show that as a whole the amount of water vapour does not change significantly over the course of the day. Perhaps the more important fact is that the phase of the diurnal cycle is such that the maximum amount of water vapour peaks around 20hrs LST and then decreases to reach a minimum at 8hrs the next morning. This would suggest that the average nighttime profile over a month is probably a good representation of the average water vapour content and therefore a nighttime only water vapour climatology is probably representative of the average water vapour content. It is important to note that this is the case for Payerne, but would not necessarily apply to other locations.

Most water vapour trend studies of the troposphere focus on PWV measurements or surface humidity measurements. To compare with the literature, we also calculated precipitable water vapour trends using both RALMO and operational radiosonde measurements and found statistically significant trends of $1.3 \pm 1.53$ mm/decade (8.8%/decade, 90% significant) and $2.3 \pm 1.51$ mm/decade (14.4%/decade, 95% significant). These trends are larger than those calculated previously in Switzerland by Morland et al. (2009) and Hocke et al. (2011). Morland et al. (2009)'s nighttime trends were all positive, ranging between 0.3 and 0.9 mm/decade with uncertainties between 0.3 and 0.6 mm/decade depending on the instrument. Morland et al. (2009)'s midnight radiosonde trend using measurements from the surface to 200 hPa was $0.87 \pm 0.46$ mm/decade and was statistically significant at 90%. Hocke et al. (2011) found no trends in water vapour using microwave radiometer PWV measurements from 2004–2009. Nyeki et al. (2019) measured temperature and PWV trends from four locations in Switzerland, including the Payerne and Jungfraujoch research stations, with GPS measurements from 1996–2015. They calculated statistically significant (90%) PWV trends at Payerne of 1.03 mm/decade during cloud-free conditions using Sen's slope method. Interestingly, the all-sky (cloudy and clear) condition trend produced a smaller trend value of 0.80 mm/decade (90% significance). We also saw a similar drop in trend magnitude when including all-sky conditions in the radiosonde PWV trend with a change in trend value from $1.34 \pm 1.51$ mm/decade to $1.08 \pm 1.05$ mm/decade. Bernet et al. (2019) has published an update to the Nyeki et al. (2019) GNSS PWV trends and the Morland et al. (2009) TROWARA PWV trends using measurements from 1995–2018. They used Payerne GNSS measurements to calculate a PWV trend of $1.14 \pm 0.93$ mm/decade or $7.3 \pm 5.3$ %/decade which is statistically significant at 95%. Their GNSS trend used all-weather and both night and day measurements. Their PWV trend calculation is consistent with the RALMO and radiosonde trends at the 1-sigma level.

We have discussed the clear sky bias in our lidar PWV trends and found no statistically significant difference between the trends using all-weather measurements and our clear-sky only PWV trends. The nighttime bias in the RALMO dataset must also be discussed in comparison with the other literature and our radiosonde trends. We have calculated a statistically significant daytime-only PWV trend using the operational radiosondes with a magnitude of $1.19 \pm 0.99$ mm/decade. We also calculated a mixed PWV trend using both daytime and nighttime measurements which was also $1.18 \pm 0.97$ mm/decade. The nighttime only radiosonde trend is 0.1 mm/decade smaller than the daytime only trend and the RALMO trend is 0.14 mm/decade larger than the daytime trend. The GNSS trend calculated by Bernet et al. (2019) uses both daytime and nighttime measurements and agrees with our radiosonde trend which uses both day and night measurements. Bernet et al. (2019)'s trend is also not statistically significant from the RALMO PWV trend at the 1-sigma level. Based on these results we do not detect a bias using only nighttime measurements. While there may be a difference in the daytime and nighttime trends it would be difficult to detect with such a short time series. Hocke et al. (2017) calculated an average PWV diurnal cycle amplitude of 2% for Switzerland. Therefore, it would probably take a very long time series with very low scatter in the residuals to significantly detect a difference between day and night trends.

The PWV trends over the last 20 years have been increasing. One reason for the increase in the trends could be caused by a corresponding increase in the surface temperature trend over the same time period. We calculated a surface temperature trend with the Payerne daily radiosondes measurements from 2009 - 2019 of $1.38 \pm 1.41\,°C$ per decade. The MeteoSwiss 2017 Klimareport (MS2017, Bundesamt für Meteorologie und Klimatologie (2017)) reported an average surface temperature

trend over Switzerland of 0.34 °C/decade from 1961 - 2017, which was strongly significant. However, a steeper increase in temperature can clearly be seen in Fig. 4.1 of MS2017 for the period of 2009 - 2017, which would correspond with our larger trend value. Additionally, the MS2017 trend is for all of Switzerland, not just for Payerne. Nyeki et al. (2019) reports a surface temperature trend of 0.79°C/decade (90% significant) during cloud-free conditions and a trend of 0.59°C/decade (90%

significant) during all-sky conditions at Payerne using measurements from 1996–2015. The last 4 years have included 3 of the warmest years on record for Europe, which may have contributed to the increase in our temperature trends with respect to Nyeki et al.'s (Copernicus Climate Change Service, 2019).

Brocard et al. (2013a) conducted a detailed study of height-resolved temperature trends and found several break points in Payerne temperature trends. While the temperature in the troposphere increased by 0.64 degrees from 1980 to 2000, the

temperature trend decreased to almost zero from 2000-2010. Therefore, measurements which include the time period from 2000–2010 will likely have smaller trend values. The temperature trend appears to have now increased again after 2010, but it is not possible with our measurements to determine the most recent break point. Despite the difference in our temperature trends from the previous literature, when the PWV trends are measured in terms of the temperature trend they agree with the dependence of saturation vapour pressure with temperature that water vapour should increase by roughly 7% for every degree.

Therefore, the current PWV trends measured by RALMO and the operational radiosondes appear reasonable.

There are very few studies of height-resolved water vapour trends over the last few decades. Most tropospheric water vapour studies focus on surface or total column changes, with a couple including trends at 850 hPa (H2018, Serreze et al. (2012)). While the majority of water vapour resides in the bottom 3 km, it is still of high importance to characterize and understand water vapour's behavior throughout the troposphere. Height-resolved water vapour trends would benefit climate and forecast

models. Additionally, Raghuraman et al. (2019) suggests that different height regions in the troposphere contribute unequally to the greenhouse effect. Their study showed that the middle tropospheric water vapour contributed the most to the water vapour component in the greenhouse effect. Lidars are well-suited to height-resolved trend measurements and the authors hope that this is the first of many similar studies which will appear over the next few years. We can compare our surface specific humidity trends to the surface water vapour trends measured in Nyeki et al. (2019). They measured statistically significant

surface specific humidity trends in Payerne, during cloud-free conditions, of 0.18 and 0.23 g/kg (depending on the method). We measured surface specific humidity trends of $0.74\pm0.46$ g/kg. Both studies detected trends which agree with the assumption of conservation of relative humidity with Nyeki et al. (2019) detecting a change of 6%/°C and RALMO with $8.8 \pm 10.2$%/°C. The difference between the two surface trends is likely due to the difference in temperature trend and the time period being used; however, when converting the trends to % per °C the uncertainties in our surface trend encompass the Nyeki surface

trend value and the differences are not significant. We can also use Nyeki et al. (2019)'s Jungfraujoch trends to compare with our 700 and 600 hPa specific humidity and temperature trends. The Jungfraujoch Observatory is at 3,580 m or about 650 hPa. The Nyeki Jungfraujoch cloud-free specific humidity trend is 0.19 g/kg per decade while RALMO's trends at 700 and 600 hPa are $0.35 \pm 0.36$ and $0.22 \pm .24$ g/kg per decade. The trends are the same within their respective $1\sigma_a$ uncertainties. Interestingly, neither our trends nor Nyeki's Jungfraujoch trends conform to the hypothesis that relative humidity is conserved.

Both humidity trends with respect to temperature are at or above 14% per degree, however, they are not statistically significant

above 95% in those units. The relative humidity may be conserved over the troposphere, or at the surface, but not in individual layers. Raghuraman et al. (2019) recently tested the assumption of the conservation of relative humidity in a changing climate and found that in small geographical regions relative humidity may not be conserved over time. However, globally, relative humidity is generally conserved. Height-resolved relative humidity trends would help to determine where relative humidity is conserved in the atmosphere.

The percent variability of water vapour through the troposphere is greatest in the free troposphere. The large variability limits the ability to calculate trends in the free troposphere, as is evident by the lack of 95% significance in the trends calculated above the surface. As Whiteman et al. (2011a) and Weatherhead et al. (1998) suggested, the variability and therefore the trend residuals drive the uncertainty in the trend and ultimately determine how long is needed to calculate the trend. The noise in the residuals of the height-resolved trends varies between 11% and 24%. Given the noise at the individual levels in our height-resolved trends, detecting a statistically significant trend at 95% of 10% per decade could take between 12–24 years of measurements (Eq. 2, Whiteman et al. (2011a)). Longer years are needed at the levels with the most noise, namely between 600–400 hPa. That is not to say that one should wait that long to try and calculate a trend. The equation in Whiteman et al. (2011a) and Weatherhead et al. (1998) to estimate the number of years to calculate a trend assumes a 90% probability of a statistically significant detection (Tiao et al., 1990; Weatherhead et al., 1998). Therefore it is an upper limit on the number of years needed. It may be possible to calculate a statistically significant trend sooner, as we have done here. In fact, that equation predicts 13 years are needed for RALMO to calculate a trend at the surface at the current trend magnitude, and yet we still found a statistically significant trend. Another point worth considering is that the amount of noise may partially be determined by the vertical resolution chosen for this study. We chose to use higher resolutions, but reducing the vertical resolution of the measurements could possibly reduce the noise in the residuals thus increasing trend significance levels. We leave this experiment to future studies.

## 6   Summary and Conclusions

We used the calibrations done by Hicks-Jalali et al. (2019) (Appendix A) in conjunction with the updated water vapour OEM retrievals from Sica and Haefele (2016) to reprocess 11 and a half years of water vapour measurements from RALMO (January 2009 - August 2019). We calculated a monthly tropospheric climatology from the reprocessed measurements which shows that we have more water vapour in the summer than winter, as expected with the larger summer temperatures. The climatology also reaches lower pressures during the summer when more water vapour is present. The statistical uncertainty at the lowest pressures is 14% of the water vapour content. The total uncertainty at those pressures is on average 20% when considering the systematic uncertainties. Uncertainties at lower altitudes are dominated by the uncertainty in the calibration constant and are a constant 5% until the statistical uncertainty is larger (roughly above 350 hPa).

We also calculated the geophysical variability of water vapour in the troposphere. The geophysical variability is a selective representation since the measurement intervals in this study were restricted to relatively clear nighttime conditions with most low-level clouds and optically thick mid-level clouds removed from the reprocessing. Therefore, the variability that is presented

is representative of the nighttime structure during clear to semi-clear conditions. The same variability study was done using both daytime and nighttime operational radiosondes and no restrictions were placed on the presence of clouds or precipitation. The radiosonde and the lidar variability studies agree in the boundary layer and both datasets show that the nighttime boundary layer does not exhibit a large percent variability. The magnitude of the variability in the radiosonde data set is larger than in the lidar study in the free troposphere, but they see similar behavior in the variability. This bias is likely due to the fact that the water vapour variability measured by the radiosonde can only be estimated since their measurement uncertainties are not reported. We are likely underestimating the uncertainties in the radiosonde which would create the difference in the variabilities measured by the two instruments. In general we see higher variability in the free troposphere than in the boundary layer. Additionally, the lidar and radiosondes measure an increased variability in water vapour during the winter months.

Most previous tropospheric water vapour trend studies focus on precipitable water vapour. Therefore, we calculated PWV trends using the bootstrap method for both the radiosonde and RALMO measurements. The radiosonde trends were calculated using nighttime and daytime measurements. The nighttime PWV trends calculated by the lidar and the radiosondes were $1.32 \pm 1.51$ mm/decade (8.9% per decade) and $2.31 \pm 1.53$ mm/decade (15.5% per decade), respectively when using the same dates. When the radiosonde trend includes all available data, including those in which clouds or precipitation are present, the PWV trend decreases to $1.08 \pm 1.05$ mm/decade or 6.36% per decade. The RALMO PWV trend is statistically significant above 90%. Both radiosonde nighttime PWV trends are significant to 95%. We also compared the nighttime only trends to daytime-only PWV trends using radiosonde measurements. We measured a daytime PWV trend of $1.18 \pm 0.97$ mm/decade or 7.20%/decade which was statistically significant at 95%. When combining daytime and nighttime radiosonde measurements we detected a similar trend of $1.19 \pm 0.98$ mm/decade or 7.20%/decade. Due to the large uncertainties in the trends we cannot detect a bias between the daytime and nighttime trends at this time. These large changes in water vapour can be explained by correspondingly large temperature changes. We calculated a surface temperature trend using the measurements from the radiosonde and found an increase of 1.38°C per decade from January 2009 - August 2019. The PWV trends are then consistent with the fact that a change in 1 degree Celsius will result in a roughly 7% change in water vapour at atmospheric temperatures of around 300 K, assuming relative humidity is conserved Held and Soden (2000).

Lastly, we investigated the specific humidity trends at 11 pressure layers in the troposphere, each 20 hPa in thickness. No statistically significant trends were found at 900 and 500 hPa, or above 350 hPa. However, all other layers exhibited positive trends with statistical significance at or above 80%. The surface specific humidity trend followed the expected Clausius Clapeyron relationship and had a trend of 8.8% change in specific humidity per degree Celsius; however, when converting to % per degree the trend is no longer significant. All other trends were larger than 7% per °C, with some as large as 17% per °C (column 5 of Table 3).

We have shown that Raman water vapour lidars are useful for making detailed tropospheric water vapour measurements and can be used for long-term analysis. The water vapour concentration over Payerne is changing at twice the rate compared to trend measurements from 10 years ago. The change in precipitable water vapour is consistent with a temperature gradient of 1.38°C per decade, assuming that relative humidity is conserved. These large changes in temperature and water vapour could have concerning impacts on the climate in Payerne over the coming decades. According to Held and Soden (2006), as water

vapour increases in the atmosphere the circulation of water vapour increases and wet regions become even wetter. The increase in saturation vapour pressure with the increasing temperatures could lead to stronger precipitation events (Sherwood et al., 2010). For the Swiss plateau, MeteoSwiss detects a positive, though non-significant, trend of 1.4% per decade in precipitation for the period 1961–2017, and the IPCC reports that, in general, precipitation amounts are increasing in the latitude band from 30°N - 60°N (Bundesamt für Meteorologie und Klimatologie, 2017; Hartmann et al., 2013).

We show for the first time, height-resolved water vapour trends in the troposphere. These trends suggest that relative humidity may be conserved at the surface, but not necessarily aloft. More measurements from Payerne as well as other lidar sites of height-resolved water vapour trends are needed in order to investigate our results. As of this writing, radiosondes are not yet well enough characterized to calculate height-resolved trends with high confidence (Elliott and Gaffen, 1991; McCarthy et al., 2009; Miloshevich et al., 2009), while satellite water vapour measurements typically have low vertical resolution and high uncertainty in the troposphere. The higher variability of water vapour in the free troposphere compared to the surface and boundary layer means that longer timelines and/or measurements with low uncertainty are required to establish statistically significant trends at the 95% level.

## Appendix A:   The RALMO Calibration Time Series

Hicks-Jalali et al. (2019) discussed the trajectory calibration technique which we used to calibrate RALMO from 2011 to 2016 using GRUAN-certified sondes. This method is valuable because it allows for the drift of the sonde away from the launch site. The method does not require the use of GRUAN sondes specifically, but would work with any sonde that reports wind speed and direction, or latitude and longitude coordinates. However, the use of GRUAN-certified radiosondes is preferable because it allows the researcher to calculate uncertainty budgets for every calibration. We found that the average total uncertainty in our calibration was around 5% for both the trajectory method and the traditional techniques and the uncertainty did not increase over the time period of the calibration study. The majority of the uncertainty ( 4%) is due to the uncertainty of the radiosonde measurement. While the uncertainty of the calibration constant did not increase over the course of the study, the value of the calibration constant did increase by at least 30% over the course of the five years. It is thought that this change in the calibration constant is due to a differential aging in the nitrogen and water vapour photomultipliers (Simeonov et al., 2014).

It has been well discussed in the lidar and trends communities that abrupt changes in calibration or any discontinuities in time series should be avoided (Whiteman et al., 2011b; Weatherhead et al., 1998). Therefore, the 25 calibration nights from the Hicks-Jalali et al. (2019) study would not be enough to use for a trend analysis, as they are too sparsely distributed and do not encompass the entire time series. They also do not fully characterize the evolution of the calibration factor over the course of RALMO's lifetime. For this reason, internal calibration methods are extremely useful as they can provide a continuous calibration function (Leblanc and McDermid, 2008; Venable et al., 2011). Leblanc and McDermid (2008) found that a hybrid method of combining an internal lamp calibration and an external calibration with a radiosonde decreased their calibration uncertainty to less than 2%.

In 2014, a UV lamp was installed at RALMO to internally calibrate the lidar following the procedure outlined in Simeonov et al. (2014), however it lacked sufficient stability and was removed from the system. An alternative method called the "Background Calibration Method" (BCM) was implemented instead. The BCM uses solar background to compute the relative calibration coefficient for RALMO (Voirin, 2017). The solar background calibration method was first introduced in Sherlock et al. (1999). We will summarize and then apply the technique to the RALMO water vapour time series used in this paper.

The calibration coefficient is the constant which converts the relative lidar profile into physical units, such as mixing ratios (g/kg). When measuring with a constant light source, such as a lamp, the ratio of the signals between the nitrogen and water vapor channels is a measurement of the ratio of each channel's efficiency. Measuring the ratio over time creates the relative calibration time series $r_{solar}(t)$:

$$r_{solar}(t) = \frac{N_{H,const}(t)}{N_{N,const}(t)}, \tag{A1}$$

where $N_{X,const}(t)$ is the water vapour or nitrogen signal measured with a constant light source (either a lamp or solar background). This function can then be normalized by an external calibration measurement taken at a time $t_0$ or by normalizing using an entire time series of measurements, such that the new calibration time series becomes:

$$C(t) = C_{ext}(t_0)r_{solar}(t). \tag{A2}$$

The external calibration, $C_{ext}$), at a time $t_0$ can be chosen at any point in the time series, or be an average of several points. The solar background signal for both the nitrogen and water vapour channels can be calculated by taking the average value of the background signal above 50 km. At these altitudes in a raw 1-minute profile, the signal is completely due to background solar radiation and not photons emitted by the laser. Therefore, during the daytime, all photons should be solar photons or tube background. One must be careful to consider both the diurnal and seasonal solar cycles when using this technique, therefore, it is important to use the same solar zenith angle each day at the highest possible angle. The highest solar zenith angle on the winter solstice is $20°$. Therefore, the calibration is conducted each cloudless morning when the sun is at a solar zenith angle of $20°$. The ratio of the nitrogen solar background and the water vapour solar background were then used to calculate the $r_{solar}$ function. We do not correct the solar background time series for differential transmission between the nitrogen and water vapour channels. A sensitivity study using over three years of RALMO data was used to assess the value of $\Delta\alpha$, where $\Delta\alpha = (1 - \tau_{N_2})/(1 - \tau_{H_2O})$. The study found that the overall impact of the differential transmission correction amounts to a maximum of 5%, with an average of 2%. These findings are in agreement with what was found in the study by Whiteman and colleagues (1992). However, the GRUAN radiosonde calibrations were corrected for differential aerosol transmission (Hicks-Jalali et al. 2018), and the relative calibration time series is scaled by the GRUAN radiosonde external calibration. Scaling the calibration time series with the GRUAN radiosonde calibration creates a continuous calibration time series from 2008 until the end of 2018 (Fig. A1).

The black points in Fig. A1 are the solar background calibration time series. The background values clearly exhibit a linear component which would indicate that the nitrogen and water vapour photomultipliers are differentially aging (assuming that the aging is linear). However, there is also a more complicated behavior embedded in the time series, particularly towards the

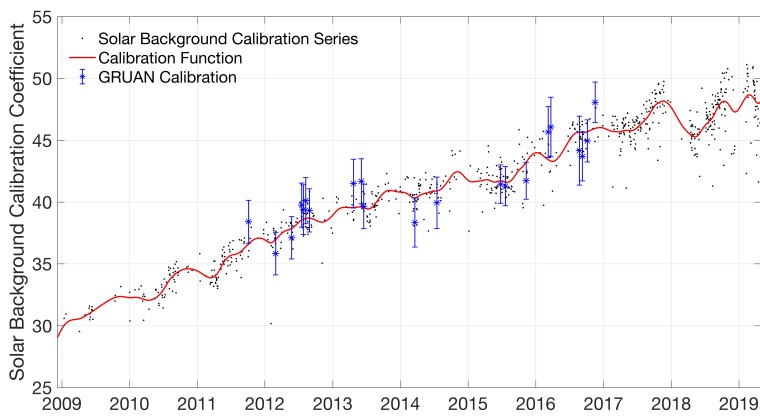

**Figure A1.** The final normalized calibration time series. Black points are the solar calibration values from January 2009 to August 2019. Blue points are the trajectory calibration values with their uncertainties. The red line is the smoothing spline fit to the solar calibration points.

end of 2017 and 2018 which would make it inappropriate to simply fit a linear function to the time series. Therefore, we chose to fit a smoothing spline to the time series, which is represented by the red line in Fig. A1. The blue points show the calibration values from Hicks-Jalali et al. (2019) with their respective uncertainties. The abrupt dip in calibration values after 2018 may be due to the installation of a new laser at that time, but the source has not yet been determined.

5    The uncertainty in the calibration function was estimated using the standard deviation of the de-trended background time series. The percent uncertainty is the standard deviation divided by the mean calibration value. It agrees with the average uncertainty of 5% from the GRUAN calibrations in Hicks-Jalali et al. (2019). While this method did not decrease the uncertainty of the calibration, it allowed the use of a continuous and consistent calibration constant for the entire water vapour time series.

*Data availability.* The water vapour climatology is available in the Zenodo database (10.5281/zenodo.3941113). Co-Author Alexander Hae-

10  fele can be contacted for access to raw RALMO and Payerne radiosonde measurements (Alexander.Haefele@meteoswiss.ch).

*Author contributions.* Shannon Hicks-Jalali is responsible for processing the RALMO and radiosonde measurements for the climatology, as well as calculating the climatology and trends. She is also responsible for normalizing the GRUAN calibration time series to the solar background time series. An earlier version of this study is part of her thesis. Robert J. Sica was responsible for writing the original OEM code. Robert J. Sica and Alexander Haefele are responsible for supervision of the thesis as well as helping with preparation of the manuscript.

15  Giovanni Martucci helped develop the solar background calibration for RALMO and provided access to the RALMO and radiosonde measurements. He also helped with manuscript preparation. Eliane Maillard Barras helped with the trend calculations. Jordan Voirin also helped develop the solar background calibration for RALMO and did the initial comparison study to the operational and GRUAN sondes.

*Competing interests.* The authors have no competing interests.

*Acknowledgements.* The authors would like to thank GRUAN for providing the corrected radiosondes. Shannon Hicks-Jalali would like to thank Dr. Ali Jalali for providing helpful advice on the process of calculating the climatology. The authors would also like to thank Valentin Simeonov for his work on the RALMO lidar. We also thank Dr. David Whiteman and our anonymous reviewer for their very valuable and thoughtful comments that improved this paper. This project has been funded in part by the National Science and Engineering Research Council of Canada through a Discovery Grant (Sica) and a CREATE award for a Training Program in Arctic Atmospheric Science (K. Strong, PI), and by MeteoSwiss (Switzerland).

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
