# Peer review of "A Raman Lidar Tropospheric Water Vapour Climatology and Height-Resolved Trend Analysis over Payerne Switzerland"

_Atmospheric Chemistry and Physics, 2019_

## Referee Comment (RC1) · Anonymous Referee #2 · 7 Feb 2020

The present work makes an important effort to build a water vapor climatology using Raman lidar measurements. Authors exploit a more than ten years database recorded at Payerne, Switzerland and very carefully distract the uncertainty, in order to estimate the natural variability in each month and atmospheric layer as long as the corresponding trends. A very detailed approach has been applied to compare the retrievals with radiosonde data.

There is the structural problem for a lidar based climatology, that there are huge gaps in the timeseries and the results are biased towards specific atmospheric conditions, which combined with the lack of long lidar time series, results in the absence of such

works in the literature. This high quality work deals with these matters and it is sufficient clear to the reader how this study should be interpreted.

Basic comment.

Major disadvantage for a climatology study is the limitation to the nighttime and cloud-less conditions. What I miss in the conclusions is who from the scientific community can use and benefit from such a climatology of nighttime water vapor.

P2l17-18 . To my understanding, the variation could be a lot more than 100%, depending on meteorological conditions. I suggest to add a reference for this number, or restate the sentence in a more general way.

P3 l8-9. "published Raman wv lidars" - "publications on Raman wv lidars"... Also, Goldsmith et al., 1994 is not in the last decade.

P3 l25. I think it should be highlighted that this trend is inside the uncertainty range.

P4 l22-25 It is important to make at least a short summary here (few sentences) for the method used for the retrieval, because it is crucial for understanding the rest of the manuscript.

P4. L18 I think, it should be added a summary and discussion about the stability of calibration and any issues rising from it.

P4 l 29. The abbreviation OEM is nowhere defined and it is not well known. Also, it should be explained this approach , why it was selected and any drawbacks that it causes.

P5 l7 This fact should be explained and provide some reference.

P5 l22 Should the 30 min be continuous? If not, could natural variability add noise, when it could be hours apart?

P5l35 It seems that the OEM output is one profile per night. But it is nowhere clearly

stated. Is it possibly to discuss the variations expected if treated differently, on a clear winter night that could last 15 hours?

P7 l 23 It is not clear what a "cost threshold of 3.5" is.

P8 l5 Are these the uncertainties discussed in the next page? If so, I suggest moving this plot and paragraph, after the definition of the uncertainties.

P8 l7 The fact that the highest uncertainties are associated with the very low concentrations in the upper troposphere should be discussed here and probably add some examples or a plot of absolute range of specific humidity uncertainty at different levels.

P10. L9. I think it is more reasonable to integrate the radiosonde starting from 100m (or the height that lidar measurements are trustworthy) , and compare this modified PWV, if the full profiles of the radiosonde are available. By this approach the results are directly comparable and not affected by any bias for the near surface area.

Figure 5. Only the days with lidar profiles are used. But as discussed earlier, it could be that the lidar profile is constructed from measurements hours away from the radiosonde, during cloudy nights. I suggest to investigate if using only profiles with data close to radiosonde timestamp could lower the biases.

Figure 6 It is not easy to claim that the natural variability at 230-250hpa is 80-90%, where the absolute values suggest is almost no humidity and the uncertainties are very high.

P12 l3. The high variability in the 600-400hpa region is one of the most interesting findings of the study. Figure 7 adds a lot of credibility to this pattern. I was wondering if a similar Temperature variation plot could provide more information for this behavior.

Îd'able 1 It is not clear how the trend %/C° is calculated for RALMO.

Table 2, specify that these trends are derived from RALMO.

Summary and Conclusions sections are overlapping. I suggest to merge in one section.

P20 l12, EOS climatology refers to what region?

Congratulations on the very interesting work

---

## Referee Comment (RC2) · David Whiteman (Referee) · 23 Feb 2020

Review of Hicks-Jalali et al. "A Raman Lidar Tropospheric Water Vapour Climatology and Height-Resolved Trend Analysis over Payerne Switzerland" by David N. Whiteman

The subject paper makes use of a 11.5-year time series of measurements from the RALMO Raman Lidar in Payerne Switzerland to calculate a nighttime water vapor climatology and height-resolved trends. The authors use the interesting new Optimal Estimation Method technique for Raman water vapor lidar retrievals introduced by Sica et al. a few years ago. The work is welcomed as extended data sets from Raman lidar have been under utilized in the past for this kind of analysis and I encourage the authors to continue this valuable work. I have a few major concerns and several minor ones and I look forward to seeing the revised manuscript.

Major Concerns

1.  It is clear that to calculate PWV and trends as a function of altitude that nighttime clear weather profiles from the Raman lidar need to be used. What is the significance of limiting this trend study to nighttime only results? This question is not dealt with much in the text yet the radiosonde data are used to derive nighttime trends as well, if I understand correctly. It would be very instructive to use the radiosondes for trend calculations using daytime data only to be this nighttime limited lidar study in more context. Those results could be used to address the question of nighttime bias.
2.  The authors also study geophysical variability including in the boundary layer in section 3.1. The value of such a nighttime only study of variability should be justified and, if possible, contrasted with the values in the daytime. The variability the authors have calculated, particularly in the BL, will likely be biased by the lack of daytime measurements. Why not include all day and night lidar data in a variability study which hopefully would at least be valid through the BL. Then the nighttime only results can be contrasted with that. Also, the authors use the radiosonde data to compare nighttime variability with the lidar. Why not include daytime radiosondes as well in the variability study to put the nighttime limited lidar-based study in more context?
3.  The authors reference Whiteman et al., 2011 as indicating that their 11.5 year data set should be sufficient for revealing trends. However, the results in Whiteman et al. indicated that trends could be resolved at the 200 hPa level (approximately the most efficient level for trend detection according to the Whiteman study) if noise-free measurements were available 30x per month. The RALMO 30-minute profile measurements, according to figure 1, are available usually less than 15x per month. So the Whiteman results would seem to indicate that trend detection with the RALMO dataset would often fail and in fact from the authors results in Table 2 most trends presented are not revealed at the 95% confidence level, which seems consistent with the Whiteman et al. results.
4.  The authors refer to Table 1 of Weatherhead et al., 1998 and seem to be referring to the $\sigma_N$ term as the "uncertainty of the measurement". Instead this term represents the standard deviation of the noise of the time series, which is comprised both of natural atmospheric variability and measurement variability due both to random and systematic sources. This number can be calculated from the time series itself as it seems the authors do at other places in the manuscript. In Whiteman et al. 2011, using radiosonde data, values of $\sigma_N$ that ranged between ~20-80% were found. I would expect values of $\sigma_N$ much larger than the 6% used here.

Minor Concerns

1. Occurs in several places. The terms water vapor mixing ratio and specific humidity seem at times to be used interchangeably. They are different quantities so please be clear on what quantity is calculated and maintain consistency.
2. P 2, line 18. Suggest change to "Due to this huge variation, quantifying water vapor trends requires ..."
3. P2, line 32. AIRS is referred to as a "satellite". It is an instrument on the Aqua satellite.
4. P2, lines 32-33. Authors state that AIRS can measure down to the surface. The reference provided discusses a vertical resolution of AIRS of approximately 3 km in the lower troposphere. I do not believe that a measurement with vertical resolution of ~3km can be considered to extend to the surface. Please revise this claim to be more consistent with your next sentence which seems to contradict this one by stating that the vertical resolution of satellite measurements is "typically on the order of kilometers".
5. P3, line 5. Suggest change to "...trend measurements are ..."
6. P3, line 9. Complete the thought with, e.g. " ...have been run operationally over the last decade (refs) but none has been used to support a study of trends as done here." or something to that effect.
7. P3, line 12. "...many of which were insignificant." implies that some were significant. Please expand briefly to discuss the details of the trends that were found to be significant.
8. P3, line 20. Suggest change to "...found a positive but insignificant trend ..."
9. P3 lines 25-26. The PWV trend values from Cezeaux have already been mentioned above. Delete this redundant information.
10. P3, line 29. A 20-year old reference (Weckwerth et al, 1999) is used to support claim about routinely available measurements with resolution better than 1 km. Can you find a current reference that supports that claim? The reference below may help
    1. https://agupubs.onlinelibrary.wiley.com/doi/abs/10.1002/2014RG000476
11. P3 line 34. This study is described here as being 10 years in length but elsewhere 11.5 years is used. Please reconcile.
12. P3. Line 35. This is the place to first note that this study uses only nighttime measurements.
13. P4 line 14. Suggest change to "...laser operating at 30 Hz..."
14. P4, line 33. Please also consider the fully propagated uncertainty estimates from the MOHAVE 2009 campaign contained in Whiteman et al., AMT, 2012, Appendix 3.
15. P 5, line 15. unit is used of "counts/bin/m". I suspect this should be "counts/bin/min" instead. Right?
16. P5, line 21. Suggest change to "...water vapor profiles have ..."
17. P5, line 26. Authors state that measurements are "naturally biased towards high pressure system conditions ...". I hope that the authors add an analysis of their daytime radiosonde data (major concern #1) and can then include a section that deals with the possibility of nighttime bias. That section could be referred to here.
18. P6, line 12. Suggest change to "Measurements from GCOS … highest quality radiosonde data product available".
19. P6, line 13. Suggest change to "Unique to GRUAN radiosonde data products is the calculation of absolute uncertainty estimates for their measurements as a function …"
20. P7, line 5. Please include some more on these non-GRUAN sondes. What type? References of previous use of this type of sonde? Efforts to get GRUAN certified?
21. P7, line 22. Authors refer to "...which pass the cost threshold of 3.5 are ...". No cost function has been introduced. Either change or add material prior to explain what this refers to.

22. P7, line 26. Suggest change to "This means that the minimum ..."
23. P9, Fig 3. Caption states fractional uncertainties but figure is labelled in %. Please reconcile.
24. P9, line 1. Fig 4 is described as being in units of mixing ratio while Fig 2 was in units of specific humidity. Did you really change units to do these calculations?
25. P9, line 10. Change "ration" to "ratio".
26. P9, line 20. I am surprised that cloud retrievals contribute much to the statistics since the laser will be attenuated quickly in clouds. So do you mean to refer here to thin or partial clouds? Also, in your retrieval, if you detect a persistent cloud why not set the Angstrom coefficient=0 (clouds are white)?
27. P 10, Fig 4. This may get too much into the details of OEM for a discussion here, but I don't understand how the calibration "constant" can have a height dependent systematic uncertainty.
28. P 10 , Fig 4 caption. Authors state that "All other uncertainties contribute less than 0.1% on average. This does not appear to be the case for the NCEP air density. Please check.
29. P 10, line 9. Is the difference in PWVs shown in Fig 5 consistent with the PWV contained in the bottom 100m? It might be a good idea to include an extrapolation of the lidar profile to the surface (using a measured surface value) to account for this missing part to try and resolve this difference.
30. P11, line 15. "The see ..." ??
31. P 12, lines 3-4. Authors state "The first, and most straightforward, explanation for the high variability at these levels is the presence of mid-level layers of clouds or aerosols". If this is the case, then you are not quantifying water vapor variability but rather something that is contaminating those calculations. Please consider whether there are additional software filters you can put on the data to prevent this contamination.
32. P13, line 7. Suggest change to "...in the free troposphere could explain ..." since you are speculating here.
33. P13, line 11. Authors state "The smaller average concentrations of water vapour in the winter leads to a larger percent variability." Smaller average concentrations do not by themselves necessitate larger percent variability. You still need to invoke some dynamical argument here to explain it.
34. P 14. line 28. "We linearly interpolated ...", This seems a curious technique although any technique that you use to reconstruct data can be criticized. In any case, it does not make sense to me to perform a linear interpolation to re-construct missing data. Your fits clearly show that some sinusoidal behavior is more appropriate and that these linearly interpolated values then look like outliers. I suggest that you perform the seasonal fit (necessarily excluding the missing data) and then use the derived seasonal function to characterize the pdf of the noise around this fit function. The seasonal fit function and the calculated distribution function can then be used to create randomize fill values for the missing data. This bootstrap technique is preferable to what the authors have done and I believe a the standard technique for dealing with missing data in these type of trend calculations.
35. P 15, Fig 8. Suggest you add the equations of the linear trend fit to the figure for each of a, b, c.
36. P 15, line 9. "The difference between the two methods represents the bias from using only semi-clear nights during clement weather." The difference seems significant at the 90% level. Given that, can you conjecture as to why use of semi-clear nights during clement weather may yield different trend values? Or is the significance in the difference in the trends not large enough to draw such conclusions? Again, it would be nice to have some results that contrast day and night values as from your radiosonde dataset.
37. P15, line 13. The RALMO calculated PWV trend is not significantly different from 0 at the 95% confidence level. Right?

38. P 16, line 14. Again linear interpolation is used here. I strongly suggest that you use the bootstrapping technique described in 34 above to fill these values. Linear interpolation will at least slightly skew the results as can be seen from your Fig 9.

39. P19, line 23. Authors give calculated trend values of 1.3 and 2.3. Please add uncertainties to these values.

40. P20, line 5. "RALMO is the only lidar ..." Be careful … the DOE ARM Raman Lidar certainly has produced such a dataset as well (over a longer time period, actually) with a higher percentage of up time.

41. P20, line 23. "Most satellite climatologies of water vapour only extend down to 300 hPa ...". I would modify this statement since AIRS and the other hyperspectral sounders (CrIS, IASI) have some lower tropospheric sensitivity.

42. P2, paragraph starting with line 30.
    1. A comparison of results of trend calculation is given here but there are no uncertainties given with any of the values. Many of the comparisons made may not be significant if you consider the uncertainties in the trend values stated. You many want to consult the 2011 Immler paper on GRUAN uncertainties for a description of language to use when describing the differences in numbers. See Table 1 for metrics to determine the use of terms such as "consistent", "in agreement", "significantly different", etc.
    2. Authors discuss both nighttime limited and day and night results here. As mentioned in major concern #1, I strongly suggest you consider expanding those daytime results to include trends calculated using your own radiosonde dataset to be able to expand this discussion. Such results seem conspicuously lacking here.
    3. Authors state in conclusion at the end of this paragraph: "Therefore, while there certainly is a natural selection bias due to only using 10 nighttime measurements in our study, the magnitude of the nighttime bias is not currently detectable." I would revisit this statement after considering the uncertainties as mentioned above and adding in results from your daytime radiosondes. It may be that the results still do not reveal a nighttime bias. However, it sounds too bold to claim that the "nighttime bias is not currently detectable." Instead a statement that seems defensible might be "Based on these results we do not detect a bias using only nighttime measurements."

43. P21, paragraph starting with line 30. Authors consider the magnitude of trends and the sensitivity of RH to changes in temperature and then compare those sensitivity numbers. Again, here, the uncertainties have not been considered. Please consider the uncertainty in both the trend of water vapor and the trend of temperature and propagate those uncertainties into your calculation of the sensitivity factors (%/C). I suspect that some of the differences are not statistically significant. And again when considering the differences you can use the language of Immler et al., 2011.

44. P22, line 11. Suggest change to "Interestingly, neither our trends nor … conforms to ..."

45. P23, line 25. Change to "ratios".

46. P24, line 6. I guess I don't understand what you're doing as it sounds like you are able to perform this check only 1 time per year (at highest SZA on winter solstice). Please expand to clarify this point.

47. P24, line 10. Referring to the black points in Fig A1 (solar background time series) … are these values corrected for differential aerosol transmission? If so please state, if not what is the magnitude of this effect?

---

## Author Comment (AC1) · 8 May 2020

**Response to Review #1:**

*Major disadvantage for a climatology study is the limitation to the nighttime and cloud- less conditions. What I miss in the conclusions is who from the scientific community can use and benefit from such a climatology of nighttime water vapor.*

At the request of Dr. David Whiteman's review, and based on your comment above, we have extended the conclusions and discussion to address the daytime and nighttime bias. We found that there was a difference of 0.2 mm/decade between the daytime and nighttime trends using the radiosondes. The difference between the two trends is much smaller than the uncertainty of each of the trends, therefore, it is not possible to say that there is a significant difference between the two regimes. We would argue that, in this case, the nighttime trend is representative of the average water vapour trend. If one had trend calculations based on a much longer time series and a smaller uncertainty, that assertion might not hold if the difference between the daytime and nighttime trend was detectable.

The PWV diurnal water vapour cycle for the Bern and Payerne was calculated in Morland et al. (2009) and most recently in Hocke et al. (2017) using microwave radiometer and GPS measurements. The average PWV diurnal amplitude is 0.5 mm, with a minimum of 0.1 mm and a maximum amplitude of 0.7 mm depending on the season (lower amplitudes in the winter and higher in the summer). Hocke et al. (2017) found that this translated to an average of 2% change in water vapour content over the course of the day with respect to the daily mean. More importantly, the peak of the diurnal cycle occurs around 19-20h local time and then reaches a minimum around 8-10h local time. This would suggest that the average nightly profile is actually a good indicator of the daily mean given the phase of the daily cycle of the region. Therefore, in this case, using the nighttime water vapour climatology can be assumed to be a good representation of the average water vapour. It is important to note that the same might not be true of other regions.

We have added a significant amount of text discussing the daytime and nighttime bias as well as the usefulness of the nighttime only climatology to the discussion section. We refer you to the new version of the paper for the new text.

P2 I17-18: *To my understanding, the variation could be a lot more than 100%, depending on meteorological conditions. I suggest to add a reference for this number, or restate the sentence in a more general way.*

You are right, a better phrasing would be that the "water vapour content can change by more than 100% over the course of the day".

The new sentence is as follows:

"Measuring an atmospheric water vapour trend … troposphere can change by more than 100% on a daily basis."

P3 l8-9: *"published Raman wv lidars"* - *"publications on Raman wv lidars"... Also, Goldsmith et al., 1994 is not in the last decade.*

We agree that "publications on" would be a better phrasing. The sentence has been changed to:

"As far as we are aware there have been only four publications on operational Raman water vapour lidars in the last 2 decades (Goldsmith … etc)"

P3 l25: *I think it should be highlighted that this trend is inside the uncertainty range.*

The sentence will be changed to:
"They found a positive, but statistically insignificant, PWV trend at Cezeaux using …"

P4 l22-25: *It is important to make at least a short summary here (few sentences) for the method used for the retrieval, because it is crucial for understanding the rest of the manuscript.*

We would be happy to add a few sentences summarizing the optimal estimation method here. We will refer the reader to Sica and Haefele (2016) as well as Rogers (2000) for details.

The following sentences have been added to the text:
The OEM uses Bayes' theorem to constrain the solution space for the retrieval. It does this by adding in the use of an *a priori* state (**x_a**). A probability of any given state of the system is assigned, assuming the errors of the system are Gaussian. The optimal solution for the system is then found by minimizing the cost of the solution, where the cost is defined as:

$$Cost = \left[\frac{1}{2}(\mathbf{y} - \mathbf{F}(\mathbf{x}, \mathbf{b}))^T \mathbf{S}_\epsilon^{-1}(\mathbf{y} - \mathbf{F}(\mathbf{x}, \mathbf{b}))\right] + \frac{1}{2}(\mathbf{x} - \mathbf{x_a})^T \mathbf{S_a}^{-1}(\mathbf{x} - \mathbf{x_a}).$$

The measurement vector is represented by **y**, **F** is the forward model for the lidar, **x** is the vector containing all retrieval parameters, **b** is the forward function parameter vector, **S_a** is the covariance matrix of the *a priori* values, and **S_e** is the measurement covariance matrix. The cost function consists of two terms. The first term is a weighted least-squares regression. The second term is a regularization term, which provides additional information to the solution through the specification of an *a priori* state. The *a priori* covariance matrix and the measurement covariance matrix define the solution space of the retrieval. Minimizing the cost function produces the retrieval solution **x**, where the solution is then the maximum *a posteriori* solution based on the probability distribution functions and is given by:

$$\hat{\mathbf{x}} = \mathbf{x_a} + (\mathbf{K}^T S_\epsilon^{-1} \mathbf{K} + S_a^{-1})^{-1} \mathbf{K}^T S_\epsilon^{-1} (\mathbf{y} - \mathbf{F}(\mathbf{x_a})) = \mathbf{x_a} + \mathbf{G}(\mathbf{y} - \mathbf{F}(\mathbf{x_a})),$$

where **K** refers to the Jacobian matrix, **G** is the gain matrix. Where the Jacobian matrix is dy/dx and the Gain matrix is dx/dy. An in depth description of OEM theory applied to atmospheric physics can be found in Rogers (2000).

*P4. L18: I think, it should be added a summary and discussion about the stability of calibration and any issues rising from it.*

We'd be happy to add a few sentences here about the calibration. The following sentences have been added:

The continuous calibration is an internal calibration technique using the ratio of the solar background between the nitrogen and water vapour channels and was first introduced in Sherlock et al. 1999. The internal solar calibration method produces a relative calibration function which is scaled to the external calibration using the GRUAN-corrected radiosondes. The calibration function has an uncertainty of 5% of the calibration value and a corresponding uncertainty of 5% in the final water vapour mixing ratio. The uncertainty in the solar calibration method is the same as what would be introduced by using GRUAN-corrected radiosondes for external calibration (Hicks-Jalali et al. 2018). However, by using an internal calibration function, the lidar trends remain mostly independent of an external instrument.

P4 l 29: *The abbreviation OEM is nowhere defined and it is not well known. Also, it should be explained this approach , why it was selected and any drawbacks that it causes.*

Our apologies for not defining it previously. It should have been defined in the previous paragraph and that has now been fixed, in addition to a brief OEM description. There are multiple papers describing the background of OEM since it became an important technique for retrievals from satellite instruments in the 1970s. The third paragraph of section 2.1 has been rewritten as follows:

"One of the advantages of using an OEM retrieval over the traditional method (Whiteman et al. 1992; Whiteman, 2003) is the addition of the uncertainty budget and averaging kernels for each profile. The addition of the averaging kernels in particular are important because they may be used to more accurately compare results with other instruments which utilize OEM retrievals, such as satellite-based limb-sounding instruments, fourier transform infrared spectrometers, or microwave radiometers.

Another advantage of using OEM for lidar measurement analysis is that the measurements do not need to be corrected before being used for the retrievals. It can be more difficult to accurately propagate uncertainties through corrections to measurements which would prevent

a complete uncertainty budget from being produced on a profile-by-profile basis. Leblanc et al. (2016) suggests a standardized method of calculating uncertainty budgets for the NDACC group, which is rigorous, but difficult to implement on a profile-by-profile basis. Additionally, corrections to the raw measurements can further induce uncertainties in the final product which may not be accounted for. Typical corrections for water vapour measurements include: accounting for photomultiplier paralysis (dead time), background noise, overlap, differential aerosol transmission, and sometimes merging multichannel measurements. The last of these can result in unknown uncertainties and biases in the water vapour, and is not necessary in OEM since the final retrieval is one profile which has been retrieved using all available measurements (Sica and Haefele, 2016). While the OEM has its advantages, some disadvantages of the method are that it is more computationally intensive than the traditional ratio method and that it is more difficult to implement. Nevertheless, a single retrieval does not take more than 30 seconds to run on an average personal laptop and the method is still quite practical for automatic and consistent processing of large datasets such as RALMO's. Longer run times occur when retrieving a large number of variables or when the bin size of the retrieved profiles are small.

We do not correct the RALMO measurements for the aforementioned possible signal effects; however, we have done some minor pre- processing before the measurements are entered into the OEM retrieval. We used nightly-integrated profiles …. "

P5 17: *This fact should be explained and provide some reference.*

This is explained in more detail in the thesis of the first author, as well as in Sica and Haefele (2016), both of which we will add as references to this sentence.

The sentence will be changed to:
"We found that these criteria effectively removed scans measured in the presence of optically thick clouds, and only left scans measured in the presence of optically thin/semi-transparent clouds such as cirrus clouds. Cirrus clouds are accounted for through the aerosol extinction retrieval in the OEM algorithm (Hicks-Jalali et al., 2019 and Sica and Haefele, 2016).

P5 l22: *Should the 30 min be continuous? If not, could natural variability add noise, when it could be hours apart?*

The 30 minutes is not required to be continuous. However, because we remove the presence of optically thick clouds we found that the signal level does not usually vary by more than a few percent over the course of the nights. The number of nights used in the study, and the monthly averaging for the final trend analysis, also helps account for occasional nights where the average nightly profile might have additional noise.

P5 l35: *It seems that the OEM output is one profile per night. But it is nowhere clearly stated. Is it possibly to discuss the variations expected if treated differently, on a clear winter night that could last 15 hours?*

Our apologies that this was not obvious. We had tried to state it on P5 l28, but the sentence is too vague. The sentence has been changed to:

"The final input profile to the OEM algorithm is a single "nightly-integrated" profile with an altitude bin size of 30 m."

The authors did not include a study on how the water vapour changes over the course of a night using OEM. However, if many points over the course of the night were included it would introduce a large autocorrelation into the trend calculation. In fact, we found that using daily values produced a large autocorrelation factor in the time series which led us to use monthly averages instead.
Additionally, the maximum retrieval height is determined by the amount of signal in the profile. Reaching the tropopause requires using nightly profiles for RALMO. We would have had to sacrifice several kilometers of altitude to do a study with higher temporal resolution.

P7 l 23 *It is not clear what a "cost threshold of 3.5" is.*

Our apologies that this wasn't clear. We have added the definition of the cost to the introduction so that this is now clearer to the reader. Costs close to 1 generally mean that the solution is well described by the model and measurements. High costs suggest, and the "high" is relative depending on the application, that the solution does not fit the model or measurements well. A cost of 3.5, for us, is on the border of when the solutions do not fit the model well and is a conservative cutoff for our retrievals.

P8 l5 *Are these the uncertainties discussed in the next page? If so, I suggest moving this plot and paragraph, after the definition of the uncertainties.*

The authors would disagree as this paragraph is about the statistical uncertainties whereas the following paragraph is about the parameter (systematic) uncertainties, that is uncertainties which are not reduced by averaging. The text and plots align with the appropriate paragraphs.

P8 l7 *The fact that the highest uncertainties are associated with the very low concen- trations in the upper troposphere should be discussed here and probably add some examples or a plot of absolute range of specific humidity uncertainty at different levels.*

We have added the following sentences discussing the uncertainties to this paragraph:

"The average statistical uncertainties are … profiles in each month. The strength of the Raman lidar signal and the statistical uncertainty is dependent on the amount of water vapour present in the atmosphere. Therefore, high statistical uncertainties are associated with low specific humidity levels. At high pressures where more water vapour is present, such as in the boundary layer, the statistical uncertainty is less than 1%. However, at lower pressures and near the tropopause where water vapour quantities are low, the statistical uncertainties reach an average of 14%. In the winter, when the air is drier, we see slightly higher uncertainties than in the summer at the same altitudes. At the surface, 82% of the profiles used in this study have statistical uncertainties of less than 5% and 96% of the profiles have statistical uncertainties less than 10%. At 250 hPa, 77% of the profiles have uncertainties lower than 20%, while 23% had uncertainties between 20 and 25%. "

The authors would prefer not to add another figure to the paper. However, we have placed the distribution of the uncertainties for the surface and at 250 hPa in the text above to give the reader a picture of the variability in the uncertainty.

*P10. L9. I think it is more reasonable to integrate the radiosonde starting from 100m (or the height that lidar measurements are trustworthy) , and compare this modified PWV, if the full profiles of the radiosonde are available. By this approach the results are directly comparable and not affected by any bias for the near surface area.*

We did not state this in the paper, but this has been taken care of. For the trends analysis (not the IWV climatology), the radiosondes were interpolated to the same altitude grid as the lidar and the IWV was calculated over the same altitude ranges. It is the authors' opinion that the bias between the radiosonde trend on the same lidar dates is due to the uncertainty in the radiosonde measurements. We know that the radiosonde humidity measurements are not of GRUAN quality and have a larger uncertainty. When the dataset is limited to 25% of the available nights, this results in a higher uncertainty in the trend and probably the larger bias that we see between the two trends. When more nights are added, for example in the radiosonde trend which uses all available nighttime dates, the bias between the radiosonde and the lidar is decreased and the trend value converges closer to the true value.

Figure 5. *Only the days with lidar profiles are used. But as discussed earlier, it could be that the lidar profile is constructed from measurements hours away from the ra- diosonde, during cloudy nights. I suggest to investigate if using only profiles with data close to radiosonde timestamp could lower the biases.*

Unfortunately, we cannot do this suggestion for two reasons. Limiting the lidar measurements to coincident measurements with the radiosonde would severely reduce the number of nights (by roughly 50%) in our time series as nights that have clouds at that time would have to be thrown out. Additionally, we would not be able to get as high into the troposphere by limiting

the lidar data to the time of the radiosonde for the reasons discussed previously.

*Figure 6 It is not easy to claim that the natural variability at 230-250hpa is 80-90%, where the absolute values suggest is almost no humidity and the uncertainties are very high.*

The uncertainty is accounted for by subtracting it from the variability. The large variability in July and August is actually between 70-80% and is real. We have verified that there are no unusual profiles inside those months that missed the filter. The large variability at those pressures is likely due to the dynamics around the tropopause.

P12 l3. *The high variability in the 600-400hpa region is one of the most interesting findings of the study. Figure 7 adds a lot of credibility to this pattern. I was wondering if a similar Temperature variation plot could provide more information for this behavior.*

We have calculated a temperature variability plot using the radiosondes as shown in Figure 1 below.

[Figure]

Figure 1: The temperature variability represented as a percentage of the mean temperature at each pressure level. The variability is calculated using daytime and nighttime radiosondes.

The temperature variability is calculated in the same way as the water vapour variability. The variability is defined as the rms of the temperature in one month, and then divided by the mean profile (in Kelvins) to produce the percent variability. Similarly to the water vapour, there is a decrease in variability during the summer months, and higher variability the rest of the year. However, there is no differentiation between the variability in the boundary layer vs the

free troposphere as there is with water vapour. Additionally, the magnitudes of the variability are very different. If the water vapour climatology presented in the paper were in units of relative humidity, we think it would make sense to include the temperature variability in the paper. However, as the water vapour units we present are the mixing ratio, we do not think entering into a discussion of temperature variability adds much value to the paper.

*Table 1: It is not clear how the trend %/C is calculated for RALMO.*

You are correct, we did not explain how the %/C trend is calculated. The trend was calculated by dividing the water vapour trend in %q/decade (column 2) by the surface temperature trend (P16 l3). We have added the following sentence to the caption in Table 1 to state how the calculation is done:

"The trend values in column 3 were calculated by dividing the water vapour trend in column 2 by the surface temperature trend."

*Table 2: specify that these trends are derived from RALMO.*

We have changed the first sentence in the caption to:

"Table of RALMO specific humidity trend calculations for each pressure layer."

*Summary and Conclusions sections are overlapping. I suggest to merge in one section.*

We have merged the two sections as you suggested.

*P20 l12, EOS climatology refers to what region?*

According to the Hadad 2018 paper, the AIRS climatology was done for a 100 km radius around Cezeaux, France. We have added this information to the paper.

*Congratulations on the very interesting work*

Thank you very much for your time spent reviewing our paper and for providing helpful and thoughtful comments.

---

## Author Comment (AC2) · 8 May 2020

**Response to Reviewer #2 - David Whiteman**

Response to Major Concerns:

1. *It is clear that to calculate PWV and trends as a function of altitude that nighttime clear weather profiles from the Raman lidar need to be used. What is the significance of limiting this trend study to nighttime only results? This question is not dealt with much in the text yet the radiosonde data are used to derive nighttime trends as well, if I understand correctly. It would be very instructive to use the radiosondes for trend calculations using daytime data only to be this nighttime limited lidar study in more context. Those results could be used to address the question of nighttime bias.*

At your suggestion, we have added the daytime PWV radiosonde trend to our results in Table 1 of this response. We have also included trends using both day and night radiosondes. It appears that there is very little difference between the daytime and nighttime PWV trends at Payerne. Additionally, the uncertainties in the trends are so large that even at 1 standard deviation we cannot accurately detect the daytime/nighttime bias. A much longer time series would be required to differentiate between the two regimes. The average PWV diurnal water vapour cycle amplitude at Payerne and over the region is about 2% (peak-to-peak 4%). The phase of the cycle is such that, on average, the peak water vapour amount occurs around 18-19hrs and decreases to reach a minimum around 6-8hrs the next morning (Hocke et al. 2017, Morland et al. 2006). This means, on average, the nighttime water vapour trends and climatology for Payerne are a good representation of the average water vapour content. Therefore, a nighttime only study is still quite useful to the water vapour community. The new radiosonde trends have been added to Table 1 in the revised manuscript.

Table 1: Precipitable water vapour (PWV) trends for RALMO and the radiosondes. The first
column is the trend value and the second column is its corresponding 2-sigma uncertainty
calculated using Eq. 5 in the new version of the paper. The first radiosonde trend is calculated
using nighttime data coincident with the lidar measurements. The second radiosonde trend is
calculated over all available nighttime dates. The third trend uses all available daytime only data
and the last trend uses all available data over day and night.

|  | PWV Trend (mm/dec) | PWV 2*sigma (mm/dec) |
|---|---|---|
| RALMO | 1.32 | 1.51 |
| RS (lidar dates, night only) | 2.31 | 1.53 |
| RS (all dates,night only) | 1.08 | 1.05 |
| RS (all dates, day and night) | 1.18 | 0.97 |
| RS (all dates, day only) | 1.19 | 0.99 |

2. *The authors also study geophysical variability including in the boundary layer in section 3.1.
   The value of such a nighttime only study of variability should be justified and, if possible,
   contrasted with the values in the daytime. The variability the authors have calculated,
   particularly in the BL, will likely be biased by the lack of daytime measurements. Why not
   include all day and night lidar data in a variability study which hopefully would at least be
   valid through the BL. Then the nighttime only results can be contrasted with that. Also, the
   authors use the radiosonde data to compare nighttime variability with the lidar. Why not
   include daytime radiosondes as well in the variability study to put the nighttime limited
   lidar-based study in more context?*

While the authors would have liked to have used the daytime RALMO measurements it was not
possible for this trend analysis. RALMO has a known daytime bias in water vapour on the order
of 10% (Dinoev et al. 2013) which is not yet fully characterized and understood. For this reason
the authors were uncomfortable using the daytime data for a trend analysis. There is currently
work under way to characterize and correct the bias and hopefully at a future date another
daytime trends paper can be published.

However, we have now included the daytime radiosondes to the variability study (Figure 1).
Adding the daytime radiosondes such that we now have the total variability (combined daytime
and nighttime radiosondes) does not significantly change the average water vapour variability
significantly. The largest changes occur on the edges of the boundary layer in the summer where
the variability is increased by 10-15% when adding daytime radiosondes, and in the winter

between 350 - 200 hPa where the variability is increased by 10 - 20% (Figure 2). Everywhere else the variability remains the same. Adding the daytime radiosondes does not change the overall behavior of water vapour's variability and the radiosonde daytime and nighttime variability still mirrors that which is seen by the lidar. It is important to keep in mind the exact differences between adding daytime radiosondes to the variability cannot be taken as absolute given that there are many uncertainties associated with daytime radiosondes (Dirksen et al. 2014) which we have not taken into account and corrected for. The results here are meant only to show that the lidar captures the same behavior and to act as a sanity check.

Figure 1. Radiosonde-measured water vapour variability using both daytime and nighttime measurements (New manuscript Fig.7)

[Figure]

Figure 2: Percent difference between new Fig 7 (Day and Night measurements) and old Fig 7 (Night only). The largest differences are at the edges of the boundary layer where adding the daytime sondes increases the variability between 5 and 15% and in January and February between 300 and 200 hPa. The large difference in April between 300 and 200hPa and at 700 hPa was due to removing 2 nighttime profiles which should have been removed previously due to a faulty sensor and bad measurements.

[Figure]

3.  *The authors reference Whiteman et al., 2011 as indicating that their 11.5 year data set should be sufficient for revealing trends. However, the results in Whiteman et al. indicated that trends could be resolved at the 200 hPa level (approximately the most efficient level for trend detection according to the Whiteman study) if noise-free measurements were available 30x per month. The RALMO 30-minute profile measurements, according to figure 1, are available usually less than 15x per month. So the Whiteman results would seem to indicate that trend detection with the RALMO dataset would often fail and in fact from the authors results in Table 2 most trends presented are not revealed at the 95% confidence level, which seems consistent with the Whiteman et al. results.*

Yes, you are correct. We should have reiterated this fact in the final conclusions. We have added text to this fact in our discussion section. As the discussion section has been significantly re-written we will not paste the text here but refer you to the new version of the paper. When we mentioned your work at the beginning it was meant to serve as a hypothesis and supposition

only. We think that these sentences are confusing to the reader and have been rewritten as follows:

"We have 11.5 years of measurements and are therefore in the realm of possible detection at the 95% level; however, whether or not we can detect trends depends on the variability of the atmosphere at the levels we choose and the magnitude of the trend itself. As we will show, the variability of the atmosphere in the free troposphere will determine our ability to detect trends at the 95% level."

4. *The authors refer to Table 1 of Weatherhead et al., 1998 and seem to be referring to the σN term as the "uncertainty of the measurement". Instead this term represents the standard deviation of the noise of the time series, which is comprised both of natural atmospheric variability and measurement variability due both to random and systematic sources. This number can be calculated from the time series itself as it seems the authors do at other places in the manuscript. In Whiteman et al. 2011, using radiosonde data, values of σN that ranged between ~20-80% were found. I would expect values of σN much larger than the 6% used here.*

Thank you for catching this mistake. We had interpreted the value correctly in our results, but did not use the correct terminology in the text. The uncertainties for our trends were calculated correctly using the standard deviation of the trend residuals. We have fixed the text in the paper to say:

"According to Weatherhead et al. (1998), trends of at least 5% per decade should be calculable in 10 years provided the standard deviation of the noise in the trend residuals is less than 6% and the autocorrelation of the trend fit residuals is less than 0.4 (Table 1 of Weatherhead et al. (1998))."

*Response to Minor Concerns:*

1. *Occurs in several places. The terms water vapor mixing ratio and specific humidity seem at times to be used interchangeably. They are different quantities so please be clear on what quantity is calculated and maintain consistency.*

In all cases, the term specific humidity is used when we have actually calculated specific humidity and not the water vapor mixing ratio. While the OEM retrieval is in units of mixing ratio, we have converted them to specific humidity for the height-resolved trend analysis and the climatology to maintain consistency with the water vapour trend literature. We have added a sentence to this effect in the second paragraph of section 2.1 where the OEM retrieval is

introduced. We realize that the part where this gets confusing is when we do not convert the uncertainty discussion to units of specific humidity. We have now added a few sentences to explain the difference when the uncertainties are first introduced. The uncertainties are not converted to units of specific humidity. The differences between the uncertainties for mixing ratio and specific humidity are negligible.

2. *P 2, line 18. Suggest change to "Due to this huge variation, quantifying water vapor trends requires ..."*

We have changed this as suggested.

3. *P2, line 32. AIRS is referred to as a "satellite". It is an instrument on the Aqua satellite.*

Thank you for catching this mistake. We have corrected it.

4. *P2, lines 32-33. Authors state that AIRS can measure down to the surface. The reference provided discusses a vertical resolution of AIRS of approximately 3 km in the lower troposphere. I do not believe that a measurement with vertical resolution of ~3km can be considered to extend to the surface. Please revise this claim to be more consistent with your next sentence which seems to contradict this one by stating that the vertical resolution of satellite measurements is "typically on the order of kilometers".*

Thank you for correcting this. We have changed the sentence as follows: "...  but a few like the AIRS instrument on the Aqua satellite can accurately measure through the troposphere ..."

5. *P3, line 5. Suggest change to "...trend measurements are ..."*

We have corrected this thank you.

6. *P3, line 9. Complete the thought with, e.g. " ...have been run operationally over the last decade      (refs) but none has been used to support a study of trends as done here." or something to that effect.*

Thank you, we have changed the text to: "As far as we are aware, there have been only four publications of operational Raman water vapour lidars in the last 2 decades \citep{Goldsmith1994, Dinoev2013, Hadad2018,Reichardt2012}. While these lidars have been run operational over the last 2 decades, RALMO is the only one which has presented a water

vapour trends study that we show here."

7.  *P3, line 12. "...many of which were insignificant." implies that some were significant. Please expand briefly to discuss the details of the trends that were found to be significant.*

The authors don't think it would be a good idea to describe which trends were and were not insignificant here for the whole paper as Ross et al. (2001) is a global study and that would get quite long. However, we have added an extra sentence here so the new sentences read as follows:

"Ross et al. (2001) calculated PWV trends from 1958--1995 using radiosonde measurements from the surface to 500 hPa and found highly geographically variable trends around the globe. Their trends calculated across Europe were insignificant and differed in sign depending on the location."

8.  *P3, line 20. Suggest change to "...found a positive but insignificant trend ..."*

We have changed it as suggested.

9.  *P3 lines 25-26. The PWV trend values from Cezeaux have already been mentioned above. Delete this redundant information.*

Thank you, we have removed this redundant sentence.

10. *P3, line 29. A 20-year old reference (Weckwerth et al, 1999) is used to support claim about routinely available measurements with resolution better than 1 km. Can you find a current reference that supports that claim? The reference below may help*
    *1. https://agupubs.onlinelibrary.wiley.com/doi/abs/10.1002/2014RG000476*

Thank you for providing a better reference for this statement. This paper appears to corroborate our original statement, with the only instruments with higher than 1 km vertical resolution in the troposphere being lidars and microwave radiometers (in the boundary layer). Therefore, we think the original statement still stands but we will replace the Weckwerth et al (1999) reference with Wulfmeyer et al. (2015) instead.

11. *P3 line 34. This study is described here as being 10 years in length but elsewhere 11.5 years is used. Please reconcile.*

Thank you for catching this. The correct time is 11.5 years. We have changed this everywhere in the paper.

*12. P3. Line 35. This is the place to first note that this study uses only nighttime measurements.*

We have added the modifier nighttime to the RALMO measurements in this sentence.

*13. P4 line 14. Suggest change to "...laser operating at 30 Hz..."*

The sentence has been changed as suggested.

*14. P4, line 33. Please also consider the fully propagated uncertainty estimates from the MOHAVE 2009 campaign contained in Whiteman et al., AMT, 2012, Appendix 3.*

We have added this reference in addition to mentioning Leblanc et al. (2016).

*15. P 5, line 15. unit is used of "counts/bin/m". I suspect this should be "counts/bin/min" instead. Right?*

Yes, that is a typo thank you for catching it.

*16. P5, line 21. Suggest change to "...water vapor profiles have ..."*

We have changed this as suggested.

*17. P5, line 26. Authors state that measurements are "naturally biased towards high pressure system conditions ...". I hope that the authors add an analysis of their daytime radiosonde data (major concern #1) and can then include a section that deals with the possibility of nighttime bias. That section could be referred to here.*

We have added daytime trends using the radiosonde measurements to Section 4. We have added a sentence here referring to the section.

New Sentence: "We will compare the nighttime only PWV water vapour trends to daytime radiosonde trends in Sect. 4.2 and discuss the daytime/nighttime bias."

*18. P6, line 12. Suggest change to "Measurements from GCOS ... highest quality radiosonde data product available".*

We have changed this as suggested.

*19. P6, line 13. Suggest change to "Unique to GRUAN radiosonde data products is the calculation of absolute uncertainty estimates for their measurements as a function ..."*

We have changed this as suggested.

*20. P7, line 5. Please include some more on these non-GRUAN sondes. What type? References of previous use of this type of sonde? Efforts to get GRUAN certified?*

The Swiss C30 and C50 radiosondes are the swiss-made operational radiosondes that are not GRUAN certified. There are no published validation studies describing the radiosondes, but they have been used for comparisons with the lidar in previous papers. Alexander Haefele can be contacted if the readers would like access to internal documents on the radiosondes. We have added the following text to the paper describing the individual radiosondes used for the operational radiosonde time series.

"The Meteoswiss operational radiosonde time series uses multiple radiosonde types: the SRS400 \citep{Martin2006, Morland2009, Brocard2013} from 2009--2010, SRS-C34 from 2010--Jan 2017, SRS-C50 from Feb 2017--March 2018 , and the Vaisala RS41 (Jensen et al. 2016) from March 2018 - present day. The C34 and C50 radiosondes are manufactured by MeteoLabor. The C34 uses a Sippican hygristor for humidity measurements which is quoted to have an accuracy of 2% RH (Meteolabor 2010), however, no public validation studies have been published. The C50 is the updated version of the C34, however, no public documentation exists on the C50 specifications … The scatter in the PWV measurements did not change appreciably between the different types of radiosondes, therefore we have not attempted any homogenizing between the different data sets."

*21. P7, line 22. Authors refer to "...which pass the cost threshold of 3.5 are ...". No cost function has been introduced. Either change or add material prior to explain what this refers to.*

We have added new material to Section 2.1 discussing some OEM theory and have added the cost function. The readers should now have enough information to understand. We refer you to Section 2.1 for the new text and equations regarding OEM theory. We have not included a detailed summary of OEM because we believe that there are enough references readily available for the readers to study OEM in more detail.

*22. P7, line 26. Suggest change to "This means that the minimum ..."*

We have changed this as suggested.

*23. P9, Fig 3. Caption states fractional uncertainties but figure is labelled in %. Please reconcile.*

Thank you this should say "percent uncertainties".

*24. P9, line 1. Fig 4 is described as being in units of mixing ratio while Fig 2 was in units of specific humidity. Did you really change units to do these calculations?*

We have now explained this in more detail in the text, but the OEM retrieval is in units of mass mixing ratio. However, to maintain consistency with other water vapour trend literature we have converted the final profiles to units of specific humidity. The uncertainties were not converted because these are calculated directly through OEM and the largest difference between the mass mixing ratio and specific humidity is less than 2% of mass mixing ratio.

*25. P9, line 10. Change "ration" to "ratio".*

Thank you for catching this typo.

*26. P9, line 20. I am surprised that cloud retrievals contribute much to the statistics since the laser will be attenuated quickly in clouds. So do you mean to refer here to thin or partial clouds? Also, in your retrieval, if you detect a persistent cloud why not set the Angstrom coefficient=0 (clouds are white)?*

The persistent cloud should actually be changed to say persistent *thin* cloud. These thin clouds go undetected by our cloud filter and therefore in the automated OEM retrieval process we don't know that they exist. If someone ran the OEM retrieval for each night individually, they could certainly set the Angstrom coefficient very close to zero if they detected a cloud. Setting it to actually zero would result in a singularity in the ARTS OEM algorithm and is therefore inadvisable. We decided to run the OEM with the same settings for the entire trends analysis to maintain consistency and the nights with persistent thin clouds did not affect the water vapour profile retrievals.

*27. P 10, Fig 4. This may get too much into the details of OEM for a discussion here, but I don't understand how the calibration "constant" can have a height dependent systematic uncertainty.*

The calibration uncertainty is technically "the uncertainty of the water vapour profile due to the uncertainty of the calibration constant". Therefore, the calibration parameter uncertainty is applied to the entire profile, however, the calibration constant uncertainty is in fact a constant value.

*28. P 10 , Fig 4 caption. Authors state that "All other uncertainties contribute less than 0.1% on average. This does not appear to be the case for the NCEP air density. Please check.*

The sentence is incorrect, thank you for catching the mistake. We have added this sentence to the caption:

"The average uncertainty contribution from the NCEP air density is .3%."

*29. P 10, line 9. Is the difference in PWVs shown in Fig 5 consistent with the PWV contained in the bottom 100m? It might be a good idea to include an extrapolation of the lidar profile to the surface (using a measured surface value) to account for this missing part to try and resolve this difference.*

We have double-checked these results by using the radiosondes. We could do this by interpolating the surface data, however, we felt uncomfortable assuming a straight linear relationship between the surface measurements and the lidar measurement. We compared the PWV calculated with and without the first 100 m up to the final lidar height and found that the difference is not completely explained by the 100 m difference in the altitudes used. The average water vapour content in the first 100 m is roughly 0.2 mm. There is a slight wet bias in the radiosondes compared to the lidar of around 0.4 mm. However, the difference is within the 1-sigma standard deviation of the measurements for both the lidar and the radiosondes. The difference between the two could be caused by the nightly averaging of the lidar compared to the radiosonde. Additionally, while the same nights are used for the radiosonde and lidar comparison, the lidar may not be measuring at the same time if clouds are present. If clouds are present at the time of the radiosonde launch, then the sonde could be wetter than the lidar which did not use the measurements during that time period. The average difference between the radiosondes and the lidar PWV is 5%, which is within the uncertainty of both the lidar and the radiosonde measurements. If you feel like interpolating from the surface station would be better, we can do so.

*30. P11, line 15. "The see ..." ??*

Thanks, we have removed this artifact from a previous edition.

*31. P 12, lines 3-4. Authors state "The first, and most straightforward, explanation for the high variability at these levels is the presence of mid-level layers of clouds or aerosols". If this is the case, then you are not quantifying water vapor variability but rather something that is*

*contaminating those calculations. Please consider whether there are additional software filters you can put on the data to prevent this contamination.*

That's true, we hadn't thought about it that way. We take care of any left-over mid-level layers of clouds and aerosols through the OEM's aerosol extinction retrieval. Therefore this cannot be the explanation. We will remove it from the text. This would leave the dynamics as the main contributor to the variability in the region.

*32. P13, line 7. Suggest change to "...in the free troposphere could explain ..." since you are speculating here.*

We agree and have changed this as suggested.

*33. P13, line 11. Authors state "The smaller average concentrations of water vapour in the winter leads to a larger percent variability." Smaller average concentrations do not by themselves necessitate larger percent variability. You still need to invoke some dynamical argument here to explain it.*

We agree that this argument is not correct. A more plausible argument is that the weather in central Europe during the winter and spring/fall is highly variable due to an increase in exchanges between colder and dryer Arctic air and warmer southern air.  The dynamics of the polar jetstream heavily influence the weather during this season which could contribute to the increased variability in water vapour. We have removed those sentences from the text and added the new sentences below:

"European winter weather is highly influenced by the polar jetstream. An increase in the variability of water vapour during these months could be caused by an increase in the exchange of air between the Arctic and the mid-latitudes."

*34. P 14. line 28. "We linearly interpolated ...", This seems a curious technique although any technique that you use to reconstruct data can be criticized. In any case, it does not make sense to me to perform a linear interpolation to re-construct missing data. Your fits clearly show that some sinusoidal behavior is more appropriate and that these linearly interpolated values then look like outliers. I suggest that you perform the seasonal fit (necessarily excluding the missing data) and then use the derived seasonal function to characterize the pdf of the noise around this fit function. The seasonal fit function and the calculated distribution function can then be used to create randomize fill values for the missing data. This bootstrap technique is preferable to what the authors have done and I believe a the standard technique for dealing with missing data in these type of trend calculations.*

We have tested the technique you suggested on the trends which had missing data (all lidar trends and the radiosonde trend using only coincident dates with the lidar). We found this technique had very little effect on the trend value, however, in all cases it slightly lowered the trend magnitude. It appears that linearly interpolating between missing points slightly increases the trend value; however, it allows for the calculation of the autocorrelation of the time series. As we understood it, the suggested method does not produce a final profile with filled points. Therefore, the uncertainties in the newer bootstrapping method can only be calculated using the standard deviation of the trends. Trend uncertainties derived via the distribution of the trends result in much smaller values because the autocorrelation is not taken into account leading to a false sense of significance in the trend. As such, the authors would argue that it would be better to be more conservative and use the original trend calculations in the paper given the uncertainties calculated are probably more realistic and the difference between the two methods is roughly .25 mm/dec for the PWV trends and on average about 1%/decade for the height-resolved trends. The Tables 2 and 3 below show the differences in the trends from the two methods and their uncertainties. The Table 2 is for PWV trends and Table 3 is for the height-resolved specific humidity trends. Note that the suggested bootstrap method's uncertainties are smaller than those produced by the Gardiner bootstrap method. The suggested bootstrap method's uncertainties are smaller due to the fact that only the replaced points are shuffled and not all points therefore there is less scatter in the trend values than in the Gardiner bootstrap method.

Table 2: PWV trends using the original bootstrap method with linear interpolation and the suggested bootstrap method ("Whiteman PWV Trend")

|  | PWV Trend (mm/dec) | PWV 2*sigma (mm/dec) (w/autocorrelation) | PWV 2*sigma Gardiner Bootstrap (mm/dec) | Whiteman PWV Trend (mm/dec) | Whiteman PWV 2*sigma (mm/dec) |
|---|---|---|---|---|---|
| RALMO | 1.32 | 1.51 | 1.34 | 1.17 | 0.32 |
| RS (lidar dates, night only) | 2.31 | 1.53 | 1.30 | 2.08 | 0.34 |

Table 3: Height-Resolved specific humidity trends using the original bootstrap method with linear interpolation and the suggested bootstrap method. Trend units are in percent per decade.

| Pressure (hPa) | Trend (%/dec) | 2sigma (%/dec) (w/autocorrelation) | 2sigma Gardiner Bootstrap (%/dec) | Whiteman Trend (%/dec) | Whiteman 2*sigma (%/dec) |
|---|---|---|---|---|---|
| 950 | 12.11 | 10.25 | 6.75 | 11.08 | 2.08 |
| 900 | 0.84 | 8.15 | 5.02 | 0.37 | 1.74 |
| 800 | 9.83 | 10.45 | 7.2 | 9.05 | 2.15 |
| 700 | 13.94 | 14.36 | 12.26 | 12.94 | 3.12 |
| 600 | 15.85 | 16.92 | 13.81 | 13.40 | 3.71 |
| 500 | 4.80 | 16.19 | 14.32 | 4.97 | 3.67 |
| 400 | 10.87 | 15.79 | 12.68 | 10.60 | 3.28 |
| 350 | 12.59 | 14.07 | 13.93 | 11.43 | 2.91 |
| 300 | 7.08 | 14.01 | 11.9 | 6.53 | 3.21 |
| 275 | 3.19 | 11.6 | 9.74 | 2.45 | 3.18 |
| 250 | 3.55 | 11.48 | 10.96 | 1.39 | 3.51 |

*35. P 15, Fig 8. Suggest you add the equations of the linear trend fit to the figure for each of a, b, c.*

Adding the entire equation would not fit in the figure, however we have done the same as in Figure 9 and added the trend values to the figure with their uncertainties.

*36. P 15, line 9. "The difference between the two methods represents the bias from using only semi-clear nights during clement weather." The difference seems significant at the 90% level. Given that, can you conjecture as to why use of semi-clear nights during clement weather may yield different trend values? Or is the significance in the difference in the trends not large enough to draw such conclusions? Again, it would be nice to have some results that contrast day and night values as from your radiosonde dataset.*

It does appear to be significant. However, after further discussion between the authors, and given the new results from the daytime radiosonde trends, we are now not sure that is the case. It is more likely that the large trend is artificial and caused by the fact that such a small portion of the radiosonde dataset is used (25%) in combination with the operational radiosondes' larger errors.

When using the entire dataset the trend converges to the real value as the random error is reduced by adding more radiosondes. We have added this discussion to the text and changed the sentence to "could represent the bias from using …".

New text is as follows:

"However, the bias between the two radiosonde nighttime trends is more likely caused by the number of points used in each trend and the uncertainty in the radiosonde measurements. Only roughly 25% of the possible nights are used in the trend analysis for the lidar dates, and as the radiosondes do have a larger and uncharacterized uncertainty it is more likely that the large trend value is due to a larger scatter in the radiosonde measurements. Given that the lidar trend is much closer to the radiosonde trends which use all available nights, including the radiosonde trends calculated using daytime and nighttime measurements, the large radiosonde trend using only nights consecutive with the lidar is more likely due to random error and not a difference in weather."

*37. P15, line 13. The RALMO calculated PWV trend is not significantly different from 0 at the 95% confidence level. Right?*

That is correct. It is statistically significant at 90%, which is stated in the paper.

*38. P 16, line 14. Again linear interpolation is used here. I strongly suggest that you use the bootstrapping technique described in 34 above to fill these values. Linear interpolation will at least slightly skew the results as can be seen from your Fig 9.*

We have answered this response above in Question 34. On average the difference between the trends from the two methods is about 1% per decade.

*39. P19, line 23. Authors give calculated trend values of 1.3 and 2.3. Please add uncertainties to these values.*

Thank you for pointing this out, we have added all missing uncertainties to the trends in the summary.

*40. P20, line 5. "RALMO is the only lidar ..." Be careful ... the DOE ARM Raman Lidar certainly has produced such a dataset as well (over a longer time period, actually) with a higher percentage of up time.*

Yes, the DOE ARM Raman Lidar has certainly produced a larger dataset than RALMO. However, the authors have not found a published climatology from that lidar. We have changed the sentence as follows:

"RALMO is one of a few lidars which has produced a published high vertical resolution water vapour climatology of the troposphere with 11.5 years of consistent measurements."

It's quite possible that we've missed a reference here. So if you are aware of the paper that has their climatology we'd happily include it in our references to make sure it is cited appropriately. We have looked in the ARM database as well as through other journals without any luck.

*41. P20, line 23. "Most satellite climatologies of water vapour only extend down to 300 hPa ...". I would modify this statement since AIRS and the other hyperspectral sounders (CrIS, IASI) have some lower tropospheric sensitivity.*

Thank you for this suggestion. The authors have decided to delete this paragraph as it does not seem to fit the discussion in the previous two paragraphs, nor is it really necessary. It seems odd to say that it is difficult to compare a climatology to anything but radiosondes when we compare it to the H2018 AIRS climatology in the previous paragraphs. Therefore, we have removed it.

*42. P2, paragraph starting with line 30.*

1. *A comparison of results of trend calculation is given here but there are no uncertainties given with any of the values. Many of the comparisons made may not be significant if you consider the uncertainties in the trend values stated. You many want to consult the 2011 Immler paper on GRUAN uncertainties for a description of language to use when describing the differences in numbers. See Table 1 for metrics to determine the use of terms such as "consistent", "in agreement", "significantly different", etc.*

   We agree that more detail regarding uncertainties and/or statistical significance should be included. Unfortunately, the Nyeki paper did not provide uncertainties in their trends, however, they did provide significance at the 90% level which we have now added to the discussion. We will refer to the language in the Immler paper in the updated discussion.

2. *Authors discuss both nighttime limited and day and night results here. As mentioned in major concern #1, I strongly suggest you consider expanding those daytime results to include trends calculated using your own radiosonde dataset to be able to expand this discussion. Such results seem conspicuously lacking here.*

We have added 2 new paragraphs to the discussion section regarding the day/night bias in the trends and in the climatology. As the Discussion section has changed appreciably, we would refer you to the updated section in the paper and will not paste it here.

3. *Authors state in conclusion at the end of this paragraph: "Therefore, while there certainly is a natural selection bias due to only using nighttime measurements in our study, the magnitude of the nighttime bias is not currently detectable." I would revisit this statement after considering the uncertainties as mentioned above and adding in results from your daytime radiosondes. It may be that the results still do not reveal a nighttime bias. However, it sounds too bold to claim that the "nighttime bias is not currently detectable." Instead a statement that seems defensible might be "Based on these results we do not detect a bias using only nighttime measurements."*

We have added the new daytime radiosonde trend calculations to the discussion and based on these results (presented in Table 1 after Major Concern #1) we have not detected a bias in using only nighttime measurements. We have changed our previous language to match your suggestion. We have made many changes to the Discussion section, therefore, we refer you to the updated section in the paper for the new text.

43. *P21, paragraph starting with line 30. Authors consider the magnitude of trends and the sensitivity of RH to changes in temperature and then compare those sensitivity numbers. Again, here, the uncertainties have not been considered. Please consider the uncertainty in both the trend of water vapor and the trend of temperature and propagate those uncertainties into your calculation of the sensitivity factors (%/C). I suspect that some of the differences are not statistically significant. And again when considering the differences you can use the language of Immler et al., 2011.*

We have now added uncertainties wherever possible to the discussion. Unfortunately, in the cases of Nyeki et al. (2019) and MS2017 we cannot put the exact numbers because they are not provided. We have added ours and others wherever possible. We have also added text regarding the significance of the trends in units of % per degree C. You are correct, when propagating the uncertainties they increase and the trends lose their significance.

44. *P22, line 11. Suggest change to "Interestingly, neither our trends nor ... conforms to ..."*

Thank you for catching this mistake.

45. *P23, line 25. Change to "ratios".*

We have changed this.

*46. P24, line 6. I guess I don't understand what you're doing as it sounds like you are able to perform this check only 1 time per year (at highest SZA on winter solstice). Please expand to clarify this point.*

We apologise that the text in this section wasn't clear. We can do this check every clear day. In order to minimize the seasonality component and the difference in the optical depth of the atmosphere at different SZA, we chose the highest common SZA. The highest SZA on the winter solstice is 20 degrees. Therefore, to be consistent across the entire year, the calibration is done at the time the sun reaches 20 degrees every morning. We have changed the text to make this clearer to the reader:

"One must be careful to consider both the diurnal and seasonal solar cycles when using this technique, therefore, it is important to use the same solar zenith angle each day at the highest possible angle. The highest solar zenith angle on the winter solstice is 20 degrees. Therefore, the calibration is conducted each cloudless morning when the sun is at a solar zenith angle of 20 degrees."

*47. P24, line 10. Referring to the black points in Fig A1 (solar background time series) ... are these values corrected for differential aerosol transmission? If so please state, if not what is the magnitude of this effect?*

Our apologies we thought we had added that detail to the text. The relative RALMO internal calculated calibration time series has not been directly corrected for the differential extinction. A sensitivity study was conducted assessing the value of $\Delta\alpha$ ($\Delta\alpha = (1 - \alpha_{N_2})/(1 - \alpha_{H_2O})$) over three years of RALMO data found that the overall impact of the $\tau$ correction amounts to a maximum of 5%, with an average of 2%. These findings are in agreement with what was found in the study by Whiteman and colleagues (1992). However, the GRUAN radiosonde calibrations were corrected for differential aerosol transmission (Hicks-Jalali et al. 2018), and the relative calibration time series is scaled by the GRUAN radiosonde external calibration.

---

## Author Response (AR2)

**Response: acp–2019–1089 A Raman Lidar Tropospheric Water Vapour Climatology and Height-Resolved Trend Analysis over Payerne Switzerland Shannon Hicks-Jalali et al.**

We have changed Fig 6 and 7 as requested by the Editor. Thanks for noticing that.

**Editor comment:**

Editor Decision: Publish subject to technical corrections (22 Jun 2020) by Matthias Tesche
Comments to the Author:
Dear Shannon,

thank you for carefully revising your manuscript and for providing detailed replies to the Referee comments.
I am happy to accept your work for publication in ACP. However, I'd suggest to change the colour table used in Figures 6 and 7 to the one in Figures 2 and 3. This would be more friendly towards colour-blind readers.

Regards,
Matthias